Corrected: Publisher correction    Publisher correction

# Integrating chemical and mechanical signals through dynamic coupling between cellular protrusions and pulsed ERK activation

Jr-Ming Yang[1], Sayak Bhattacharya [2], Hoku West-Foyle[3], Chien-Fu Hung [1,4], T.-C. Wu [1,4,5,6], Pablo A. Iglesias [2,3] & Chuan-Hsiang Huang [1]

The Ras-ERK signaling pathway regulates diverse cellular processes in response to environmental stimuli and contains important therapeutic targets for cancer. Recent single cell studies revealed stochastic pulses of ERK activation, the frequency of which determines functional outcomes such as cell proliferation. Here we show that ERK pulses are initiated by localized protrusive activities. Chemically and optogenetically induced protrusions trigger ERK activation through various entry points into the feedback loop involving Ras, PI3K, the cytoskeleton, and cellular adhesion. The excitability of the protrusive signaling network drives stochastic ERK activation in unstimulated cells and oscillations upon growth factor stimulation. Importantly, protrusions allow cells to sense combined signals from substrate stiffness and the growth factor. Thus, by uncovering the basis of ERK pulse generation we demonstrate how signals involved in cell growth and differentiation are regulated by dynamic protrusions that integrate chemical and mechanical inputs from the environment.

[1] Department of Pathology, Johns Hopkins Medical Institutions, Baltimore, MD 21205, USA. [2] Department of Electrical and Computer Engineering, Whiting School of Engineering, Johns Hopkins University, Baltimore, MD 21218, USA. [3] Department of Cell Biology, Johns Hopkins School of Medicine, Baltimore, MD 21205, USA. [4] Department of Oncology, Johns Hopkins Medical Institutions, Baltimore, MD 21205, USA. [5] Department of Obstetrics and Gynecology, Johns Hopkins Medical Institutions, Baltimore, MD 21205, USA. [6] Department of Molecular Microbiology and Immunology, Johns Hopkins Medical Institutions, Baltimore, MD 21205, USA. Correspondence and requests for materials should be addressed to C.-H.H. (email: chuang29@jhmi.edu)

The Ras family of small GTPases, including H-, K-, and N-Ras, are activated by RasGEFs in response to receptor tyrosine kinase (RTK) stimulation. Through their downstream effectors such as the PI3K-AKT and MAPK/ERK signaling pathways, Ras GTPases play an important roles in cell proliferation, differentiation, metabolism, motility, and other physiological processes[1,2]. The RTK-Ras-PI3K-ERK signaling network is frequently mutated across different types of human cancers[3]. Recent years have seen the development of several important anti-cancer drugs targeting this signaling network. However, issues of efficacy and resistance remain challenging, and a better mechanistic understanding is required to cope with problems associated with available therapeutics.

The cellular responses to complex environmental stimuli are governed by the spatiotemporal dynamics of signaling networks[4]. For example, EGF and NGF both trigger ERK activation in the PC12 pheochromocytoma cells. However, the transient ERK response induced by EGF leads to cell proliferation; whereas, the sustained response to NGF causes differentiation into neurons[5]. The outcomes of other signal transduction pathways such as NF-kB and p53 are similarly linked to their dynamics[4]. Due to nonlinear feedback interactions between component proteins, signaling networks often display self-organized activities, such as stochastic pulses, oscillations, and spatial pattern formation[6,7]. Recent studies showed that in single cells, ERK activation occurs as discrete pulses, the frequency of which is modulated by growth factors or cell density to determine cell cycle entry (Fig. 1a)[8–10]. Optogenetic manipulation of ERK dynamics led to altered protein phosphorylation and gene transcription[11,12].

In migrating cells, Ras-PI3K signaling activities propagate as traveling waves on the membrane to drive the actin-based cytoskeleton during protrusion formation[13–20]. These waves reflect an underlying excitability of the Ras-PI3K signaling network, which is characterized by all-or-none activation and the existence of a refractory period upon repeated stimulation. Such excitability allows for stochastic generation of protrusions in the absence of stimuli and amplification of shallow gradients of guidance cues during chemotaxis[15,16].

Although, ERK is a well-established downstream effector of Ras, during pulsatile ERK activation no corresponding change in Ras activity is visible in individual cells[9]. Intriguingly, a dominant-negative Ras mutant completely blocked ERK pulses, suggesting that a basal Ras activity is required for ERK pulse generation[9]. However, the traveling waves of Ras and PI3K activation that drive cell migration are spatially localized and might escape detection at the whole-cell level. These considerations prompted us to revisit the relationship between Ras and ERK activities at the subcellular level (Fig. 1a). Here we show that the pulsatile activation of ERK is triggered by localized protrusions, which are driven by an excitable network of Ras, PI3K, the cytoskeleton, and cellular adhesion. Functionally, the protrusions allow cells to integrate chemical and mechanical stimuli from the environment in the regulation of ERK signaling.

## Results

### Spatiotemporal coupling between ERK pulses and protrusions.
To detect Ras and PI3K activation, we used fluorescent protein (FP)-tagged RBD (Ras-binding domain) and PH-AKT (PH domain from AKT), which are recruited to the membrane upon Ras and PI3K activation, respectively (see Methods and Supplementary Figure 1a, b). Total internal reflection fluorescence (TIRF) images revealed that RBD and PH-AKT signals, normalized to that of a membrane marker Lyn, were elevated in large protrusions (Supplementary Figure 1c-d, Supplementary Movie 1, 2). The co-localization of RBD and PH-AKT on these protrusions

(Supplementary Figure 1e) is consistent with a positive feedback loop between Ras and PI3K described earlier[15,21,22]. Additionally, increased actin polymerization as detected by the biosensor LifeAct[23] was observed in those large protrusions as well as in rapid, small undulations around the cell perimeter that were labeled with LifeAct but not with PH-AKT or RBD (Supplementary Figure 1f, Supplementary Movie 3). This apparent requirement for Ras-PI3K signaling to drive the cytoskeleton in the generation of large protrusions is similar to previous observations in amoeboid cells despite the less migratory nature of the cells examined in this study[15,24].

To measure ERK activities in live cells, we used FP-tagged ERKKTR, which undergoes nucleocytoplasmic shuttling upon ERK activation[10]. Compared to FRET-based biosensors, the single-colored ERKKTR is advantageous for multiplexing with other fluorescence biosensors, and has a superior dynamic range[25]. Consistent with previous reports[8–10], we observed spontaneous ERKKTR nuclear exit events corresponding to pulses of ERK activation (Supplementary Figure 2a). As cells underwent constant morphological changes that might affect the apparent distribution of ERKKTR, we compared the cytoplasmic and nuclear fluorescence of ERKKTR to that of a GFP volume marker. The cytoplasmic to nuclear (C/N) ratio of ERKKTR with or without normalization to GFP showed peaks that corresponded to the nuclear exit events (Supplementary Figure 2a).

To investigate the relationship between Ras and ERK activities, we co-expressed RBD and ERKKTR in the same cells. Simultaneous imaging of RBD by TIRF and ERKKTR by epifluorescence microscopy revealed that ERKKTR nuclear exit events were preceded by the formation of RBD-labeled protrusions (Fig. 1b, Supplementary Movie 4; for additional examples of protrusions preceding ERKKTR nuclear exits see Supplementary Figure 2b). As PH-AKT co-localized with RBD at protrusions (Supplementary Figure 1e), the protrusions associated with nuclear exit of ERKKTR were also enriched in PH-AKT (Fig. 1c).

We carried out quantitative analysis to determine the spatiotemporal relationship between ERK and protrusions. To extract information about protrusions, we developed a frame difference method (FDM) based on calculating the pixel-by-pixel differential of biosensor intensity between consecutive frames in time-lapse images (Methods section). Temporal averaging was applied to filter out the rapid cell boundary undulations. By setting appropriate thresholds FDM allowed for effective identification and quantification of protrusions. Kymographs revealed that nuclear exits of ERKKTR were preceded by protrusions that occurred at different areas around the cell perimeter (Fig. 1d, Supplementary Movie 4). The ERK pulses varied in their magnitude, and may occur as discrete events or fused with one another. Moreover, the magnitude of ERKKTR peaks showed a strong correlation with that of protrusions (Fig. 1e). To define the temporal relationship between ERK and protrusive activities, we identified discrete, large pulses, defined as more than 20% increase in the C/N ratio of ERKKTR. Among 12 such ERK pulses detected in seven cells, 11 were preceded by large protrusions, whereas one showed diffuse but small cellular spreading. Two different analyses were carried out to determine the lag time between protrusions and ERKKTR (C/N): (1) cross-correlation between the protrusion area and ERKKTR (C/N) over time revealed a lag of 6 min (Fig. 1f); (2) the time to reach half-maximal peak for ERKKTR (C/N) was $5.02 \pm 0.75$ min (mean ± s.e.m., $n = 11$) behind that of protrusions (Supplementary Figure 2d).

As an independent confirmation of the association between protrusions and ERK activation, we imaged cells expressing the FRET-based ERK biosensor EKAR[26] by TIRF microscopy. Indeed, protrusions were accompanied by an increase in the

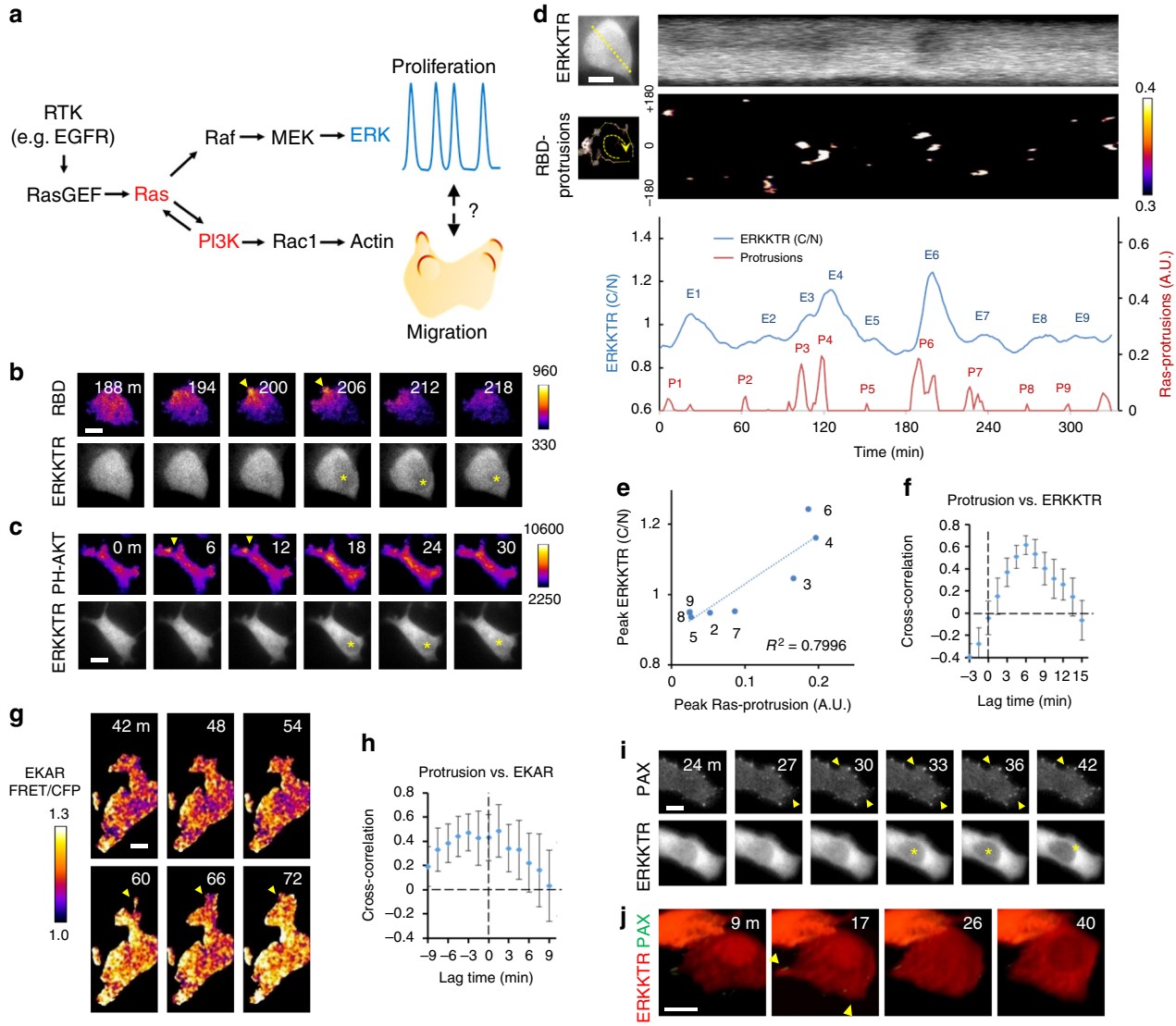

**Fig. 1** Spatiotemporal relationship between ERK pulses and protrusions. **a** The RTK-Ras-PI3K-MAPK/ERK signaling network. ERK displays pulsatile activation to drive proliferation (blue), whereas Ras-PI3K activity propagates as reaction-diffusion waves on the membrane (red) to drive the generation of protrusions during cell migration. **b**, **c** Time-lapse epifluorescence images of ERKKTR along with TIRF images of FP-tagged RBD (**b**) and PH-AKT (**c**) in MCF7 cells showing protrusions (arrowheads; color scale: fluorescence intensity (A.U.)) associated with nuclear exit of ERKKTR (asterisks). **d** Upper kymograph: temporal evolution of ERKKTR fluorescence along the dashed line across the nucleus. Lower kymograph: RBD-enriched protrusions (color scale: intensity (A.U.) identified by FDM, see Methods) around the perimeter of the same cell (corresponding to Supplementary Movie 4). Quantification of cytoplasmic to nuclear ratio (C/N) of ERKKTR (blue) vs. integrated intensity of RBD-enriched protrusions (red) over 6 h of imaging is shown below. E1–E9 mark peaks of ERKKTR (C/N); P1–P9 mark protrusive activities preceding E1–E9. **e** Plot of the magnitude of ERKKTR (C/N) peaks vs. that of RBD-enriched protrusions. The numbers correspond to the peaks in **d**. **f** Cross-correlation analysis of the lag between protrusions and ERKKTR (mean ± s.e.m., $n$ = 12). **g** TIRF images of an MCF7 cell expressing EKAR, a FRET probe for ERK activation. **h** Cross-correlation analysis of the lag between protrusions and EKAR (mean ± s.e.m., $n$ = 9). **i** An MCF7 cell expressing paxillin-GFP (TIRF) and ERKKTR-RFP (epifluorescence). **j** LLSM images of an MCF7 cell showing new paxillin-associated protrusions (arrowhead) associated with ERKKTR nuclear exit (see also Supplementary Movie 5). Scale bars: 10 µm

FRET efficiency of EKAR, indicating ERK activation (Fig. 1g; additional example in Supplementary Figure 2c). (Note that although the FRET-based EKAR can potentially reveal regional differences in ERK activity, the signal increase was diffuse across the cell area due to the rapid diffusion (diffusion coefficient ~10–100 µm²/s for cytosolic proteins) compared to the much slower kinetics of ERK activation.) Unexpectedly, cross-correlation and half-maximal time analyses revealed an absence of temporal separation between EKAR and protrusions (Fig. 1h, Supplementary Figure 2d). This discrepancy between the activation times of the two ERK biosensors relative to protrusions

can be explained by the faster kinetics of EKAR, as revealed by the responses to EGF stimulation (Supplementary Figure 2e).

As the protrusions visualized by TIRF were in close proximity to the coverslip, we speculated that they represented substrate attachment sites. Consistent with this idea, we found that new patches of paxillin, a component of the focal adhesion complex, formed on these protrusions (Fig. 1i). Using lattice light-sheet microscopy (LLSM)[27] we obtained a 4D view of the coupling between protrusions and ERK pulses, confirming substrate attachment of the ERK-associated protrusions (Fig. 1j, Supplementary Movie 5). Together these observations indicated that

ERK pulses were associated with protrusions involving coordinated activation of Ras-PI3K, the cytoskeleton, and cellular adhesion. It should be noted that cells also displayed non-coordinated activities of Ras-PI3K, actin, and adhesion that did not trigger ERK pulses. First, in the interior regions of basal cell surface we found flashes of RBD and PH-AKT activities that persisted in the presence of inhibitors for actin polymerization (Supplementary Figure 3, Supplementary Movie 6). Such activities did not cause ERK activation but interfered with the detection of correlated Ras and ERK pulses at the whole-cell level (Supplementary Figure 4, Supplementary Movie 4). Second, multiple focal adhesions preexisted in non-protruding regions around the basal cell surface and stayed unchanged throughout the ERK pulses (Fig. 1i). Third, not all cell boundary movements were followed by ERK activation. In particular, the aforementioned rapid actin-driven undulations devoid of Ras or PI3K activities (Supplementary Figure 1f) did not correlate with ERK pulses. In LLSM, the undulations appeared as rapid ruffling extending away from the substrate and were not accompanied by nuclear exit of ERKKTR (Supplementary Movie 7).

**Correlation between protrusions and ERK across cell lines**. If ERK activation is generally coupled to protrusive activities, we expect the level of ERK activity to correlate with protrusion frequencies across different cell types. To test this prediction, we examined a panel of cell lines, including MCF7 (human breast cancer), SKOV3 (human ovarian cancer), 4T1 (mouse breast cancer), HeLa (human cervical cancer), and U2OS (human osteosarcoma). The nucleocytoplasmic distribution of ERKKTR varied between individual cells of the same type and across different cell lines (Fig. 2a). Importantly, the average C/N ratio of ERKKTR was in good agreement with the level of phospho-ERK ($R^2 = 0.79$, Fig. 2b, c), thus validating the use of ERKKTR for reporting ERK activity across cell types. We analyzed protrusion frequency by FDM using images of cells expressing FP-tagged RBD or PH-AKT (Fig. 2d, e, Supplementary Movie 8). Plotting the frequency of protrusion against ERKKTR C/N ratio showed a high Pearson correlation coefficient of 0.88 between these two parameters (Fig. 2f). Thus, coupling with protrusions is a general feature of ERK activation. Whereas low frequency of protrusions drove discrete ERKKTR nuclear exit events (Fig. 1d, Supplementary Movie 4), high frequency protrusions caused persistent nuclear exclusion of ERKKTR (Fig. 2g, Supplementary Movie 9), suggesting that ERK activation lacks a refractory period characteristic of excitable systems.

**Molecular basis of the protrusion-ERK coupling**. The causative relationship between the coupled protrusions and ERK pulses was not apparent from their temporal order (Fig. 1h, Supplementary Figure 2d). Therefore, we performed perturbation analyses to probe the regulatory relationship between Ras-PI3K, the cytoskeleton, cellular adhesion, and ERK (Fig. 3a). First, genetic activation and inhibition of Ras by expressing constitutive (G12V) and dominant-negative (S17N) Ras mutants led to a corresponding change in the ERK activation and frequency of protrusions (Fig. 3b–d). Second, pharmacological inhibition of PI3K caused retraction of basal cell surface area accompanied by ERK inactivation in SKOV3 cells (Supplementary Figure 5a-c, Supplementary Movie 10). The effect of PI3K inhibition on ERK activity in MCF7 cell was less pronounced (Supplementary Figure 5d), possibly due to their lower basal ERK activity (Fig. 2). We next asked whether acute activation of Ras and PI3K can trigger ERK signaling. To this end, we used chemically induced dimerization (CID), whereby synthetic activation of a signaling pathway is achieved by membrane recruitment of an FKBP-fused

signaling activator through dimerization with the membrane-tethered FRB-Lyn upon the addition of rapamycin (Fig. 3e). While recruitment of YFP to the membrane did not activate ERK (Fig. 3f), synthetic activation of Ras through CID of YFP-CDC25, a RasGEF[28], led to the formation of paxillin-labeled protrusions followed by ERK activation (Fig. 3g, h). A similar outcome was achieved by PI3K activation using CID of iSH2, a domain from the regulatory subunit of PI3Kα [29] (Fig. 3i, j).

We next triggered protrusions by direct activation of the cytoskeleton, achieved by CID of Tiam1, a RacGEF (Fig. 3a)[29]. Strikingly, protrusions induced by CID of Tiam1 also triggered ERK activation and were labeled with paxillin (Fig. 4a, b). As an alternative approach for activating the cytoskeleton, we used PA-Rac1 to optogenetically trigger actin polymerization[30]. Upon blue light irradiation, PA-Rac1-induced protrusions were followed by nuclear exit of ERKKTR with kinetics similar to that induced by CID of Tiam1 (Fig. 4c). These results are consistent with the existence of a positive feedback from the cytoskeleton to Ras, which then activates the MAPK/ERK cascade. A similar feedback loop involving Ras, PI3K, and cytoskeleton has been suggested to regulate the migration of *Dictyostelium*[31]. In support, CID of Tiam1 also activated PI3K (Fig. 4d). Moreover, the responses of PI3K and ERK to short EGF stimulation were reduced when actin polymerization was inhibited by latrunculin (Fig. 4e, f). Interestingly, latrunculin also caused loss of paxillin patches and blocked phosphorylation of FAK (focal adhesion kinase) (Supplementary Figure 6a, b). As FAK has been suggested to mediate Ras activation by integrin[32,33], we investigated its role in protrusion-induced Ras-PI3K and ERK activation. First, inhibition of FAK reduced phospho-AKT and phospho-ERK levels (Supplementary Figure 6c, d). Second, pre-treating cells with FAK inhibitors abolished ERK activation induced by CID of Tiam1 (Fig. 4g and Supplementary Figure 6e). These observations indicate that FAK mediates protrusion-induced Ras-PI3K activation to drive ERK activation. (Note that the response of ERK to EGF stimulation was not blocked by FAK inhibition, Fig. 4g and Supplementary Figure 6e).

To see whether the pulsatile ERK activity is responsible for the dynamic generation of protrusions, we treated cells with the MEK inhibitor (MEKi) PD325901, which abolished phospho-ERK within 1 h but did not affect the level of phospho-AKT (Supplementary Figure 6f, g). Cells treated with MEKi still produced protrusions with paxillin patches upon EGF stimulation (Fig. 4h). MEKi also did not block the generation of spontaneous protrusions (Fig. 4i), which occurred at a frequency similar to that before MEKi treatment (Fig. 4j and Supplementary Movie 11). Therefore, ERK is not required for the dynamic protrusive activities.

Taken together our perturbation analyses suggest the existence of a positive feedback loop between Ras, PI3K, the cytoskeleton, and cellular adhesion that can be activated at different points to trigger ERK signaling (Fig. 3a).

**Mathematical modeling of the protrusion-ERK coupling**. Based on our findings, we propose that stochastic pulses of ERK activation are triggered by spatially localized protrusions. We built a mathematical model to capture the salient features of protrusive and ERK activities. The stochastic generation of protrusions was modeled by an excitable network composed of a self-catalyzing activator and a delayed inhibitor defined over the perimeter of a circular cell[15,34,35]. The aggregate output of the excitable network was then coupled to an ERK module that is ultrasensitive to stimuli (Fig. 5a, see Methods for details). The ultrasensitivity of the ERK module, as reported earlier[36], is supported by our observation that large ERK responses can be induced by protrusions that occupy only a small fraction of the cell perimeter

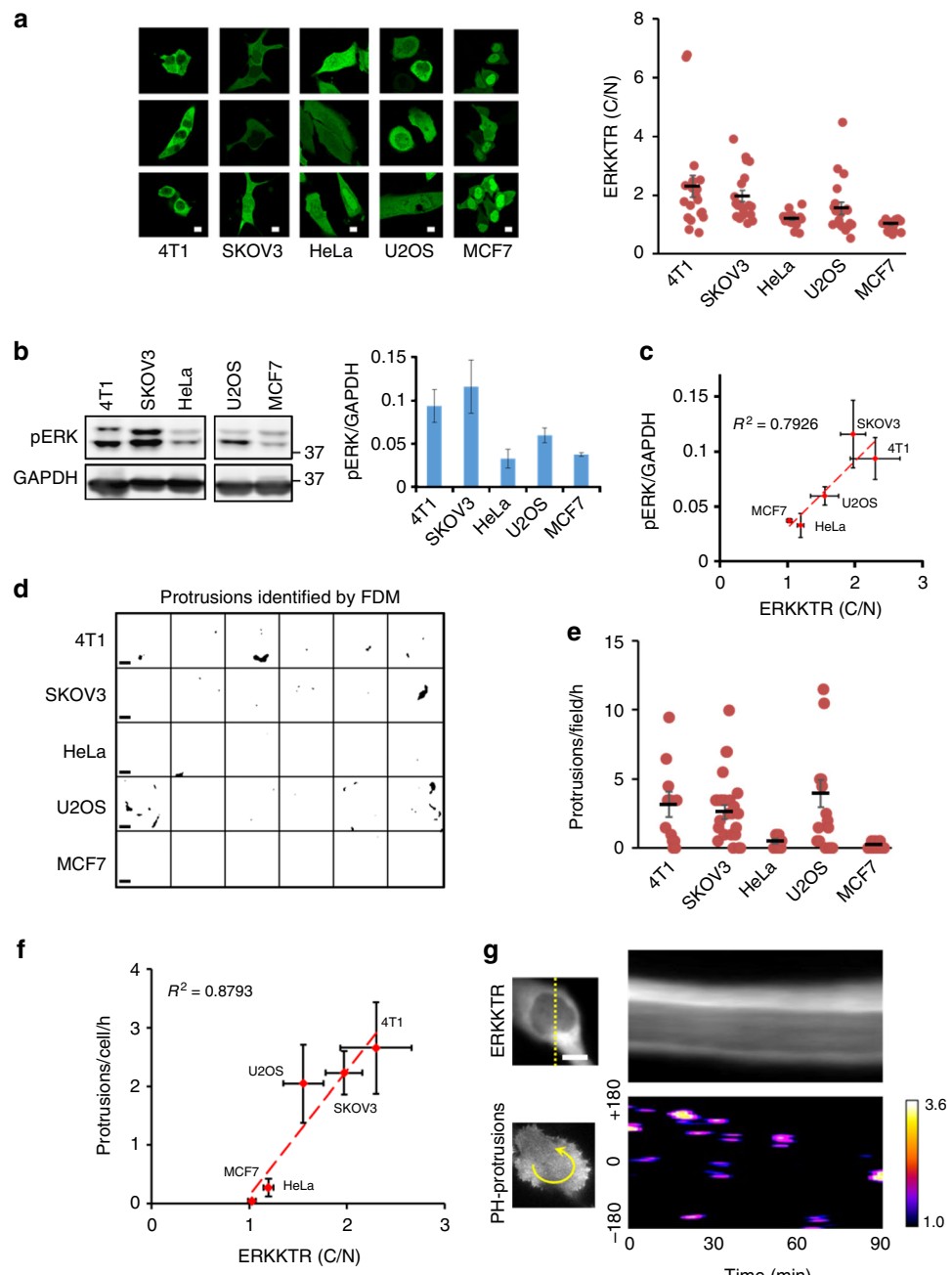

**Fig. 2** Correlation between ERK activity and protrusion frequency across cell lines. **a** Representative confocal images of ERKKTR-GFP distribution and quantification of C/N ratio of ERKKTR (mean ± s.e.m., $n = 20$ cells each) for five different cell lines. **b** Immunoblot of phospho-ERK and GAPDH and quantification (mean intensity ± s.e.m., $n = 3$ experiments; the upper and lower bands in pERK correspond to ERK1 and ERK2, respectively). **c** Correlation between phospho-ERK level and ERKKTR(C/N) across cell lines. **d**, **e** Identification and quantification of protrusions using FDM (Methods section) across cell lines. A snapshot of the FDM-processed image was shown (**d**) (see Supplementary Movie 8), along with quantification of the protrusion frequency (**e**) ($n = 13$, 33, 9, 22, 48 cells for 4T1, SKOV3, HeLa, U2OS, MCF7, respectively). **f** Correlation between protrusion frequency and ERK activity across cell lines. **g** Kymographs of ERKKTR (upper) and PH-AKT-enriched protrusions (lower, color scale: intensity (A.U.) identified by FDM, see Methods) in an SKOV3 cell showing persistent nuclear exclusion of ERKKTR throughout the imaging period (corresponding to Supplementary Movie 9). Scale bars: 10 μm

(Fig. 1d). The frequency of protrusions can be adjusted by changing the threshold of the excitable network. When the threshold was high, computational simulation of the model recapitulated the stochastic protrusive patches that drove discrete ERK pulses (Fig. 5b). Lowering the threshold generated frequent protrusions that caused sustained ERK activation (Fig. 5c) similar to the observation in cells with high ERK activities (Fig. 2g).

Our quantitative model provided an explanation for the oscillatory ERK activation following acute growth factor stimulation[37,38]. In our simulation, a sudden stimulus caused a uniform activation of the excitable network that drove a large ERK activation, which quickly shut off. After the refractory period of the excitable network passed, some regions regained responsiveness and initiated secondary firings at different locations around the cell perimeter. The secondary peaks were partially synchronized and drove repeated pulses of ERK activation with a frequency determined by the refractory period of the excitable network (Fig. 5d). When the stimulus was

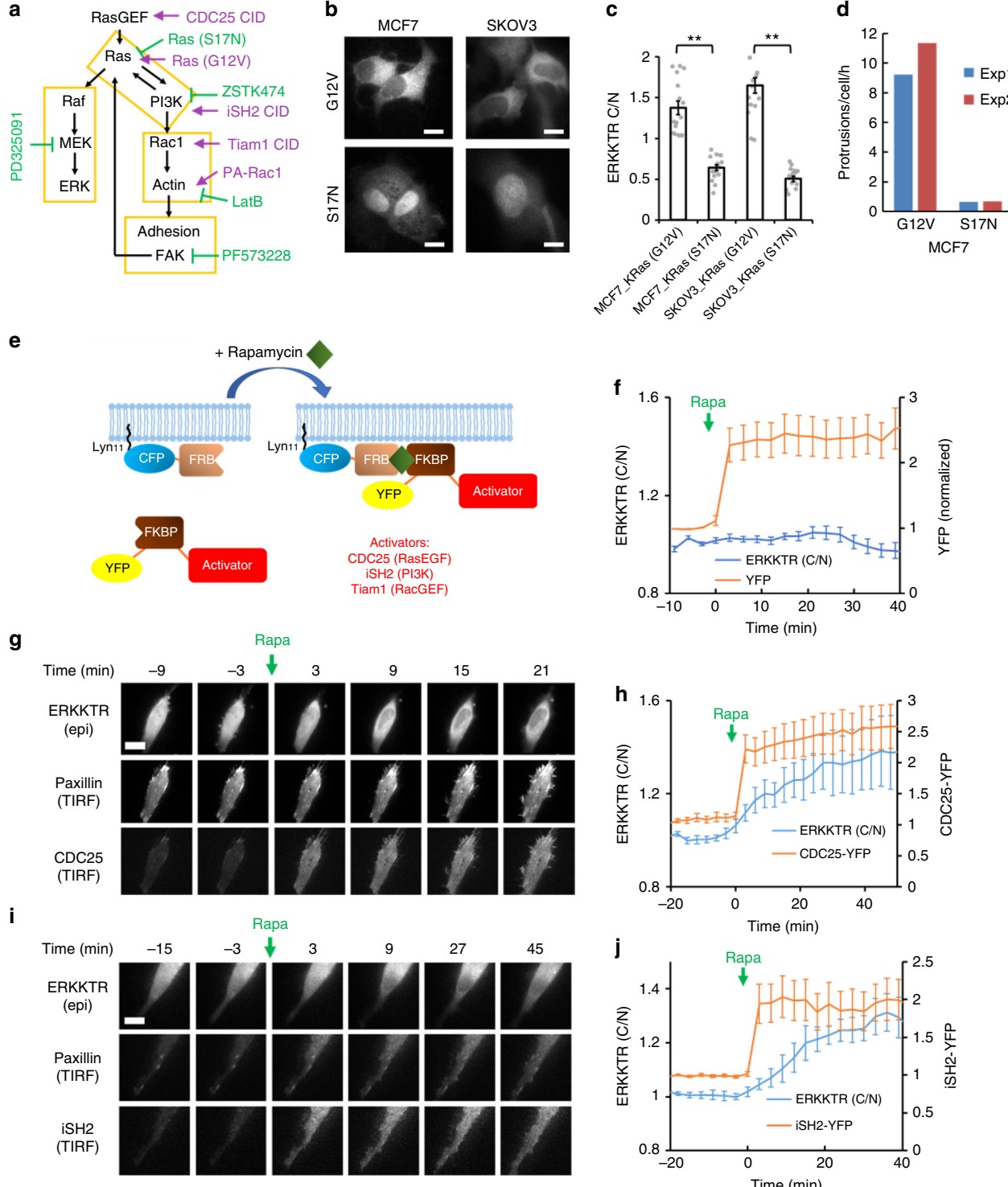

**Fig. 3** Perturbation analyses of the signaling network. **a** Schematic of activating (magenta) and inhibitory (green) perturbations to the signaling network. CID: chemically induced dimerization. **b** Representative confocal images of ERKKTR-GFP in cells expressing Ras(G12V) and Ras(S17N). **c** Quantification of C/N ratio of ERKKTR in cells expressing Ras(G12V) and Ras(S17N) (mean ± s.e.m., $n = 16$ cells). **$p < 0.001$ by two-tail unpaired $t$-test. **d** Protrusion frequency of MCF7 cells expressing Ras(G12V) and Ras(S17N). The analysis was done by FDM (see Methods) and results from two independent experiments are shown. **e** Schematic of CID. Rapamycin induces membrane recruitment of an FKBP-fused signaling activator through dimerization with the membrane-tethered FRB-Lyn, allowing for rapid activation of signaling events. **f** Recruitment of YFP-FKBP control (mean ± s.e.m., $n = 14$; intensity in TIRF normalized to pre-stimulus levels). **g** Representative time-lapse images of ERKKTR-RFP (epifluorescence), GFP-paxillin (TIRF), and YFP-FKBP-CDC25 (TIRF) showing that acute activation of Ras by CID induced adhesive protrusions and ERK activation. **h** Kinetics of YFP-FKBP-CDC25 recruitment and ERKKTR (C/N) response to rapamycin treatment (mean ± s.e.m., $n = 8$ cells). **i, j** Representative images for PI3K activation by CID of iSH2 (**i**) as well as the corresponding kinetics of YFP-FKBP-iSH2 and ERKKTR responses (mean ± s.e.m., $n = 7$ cells) (**j**). Scale bars: 10 μm

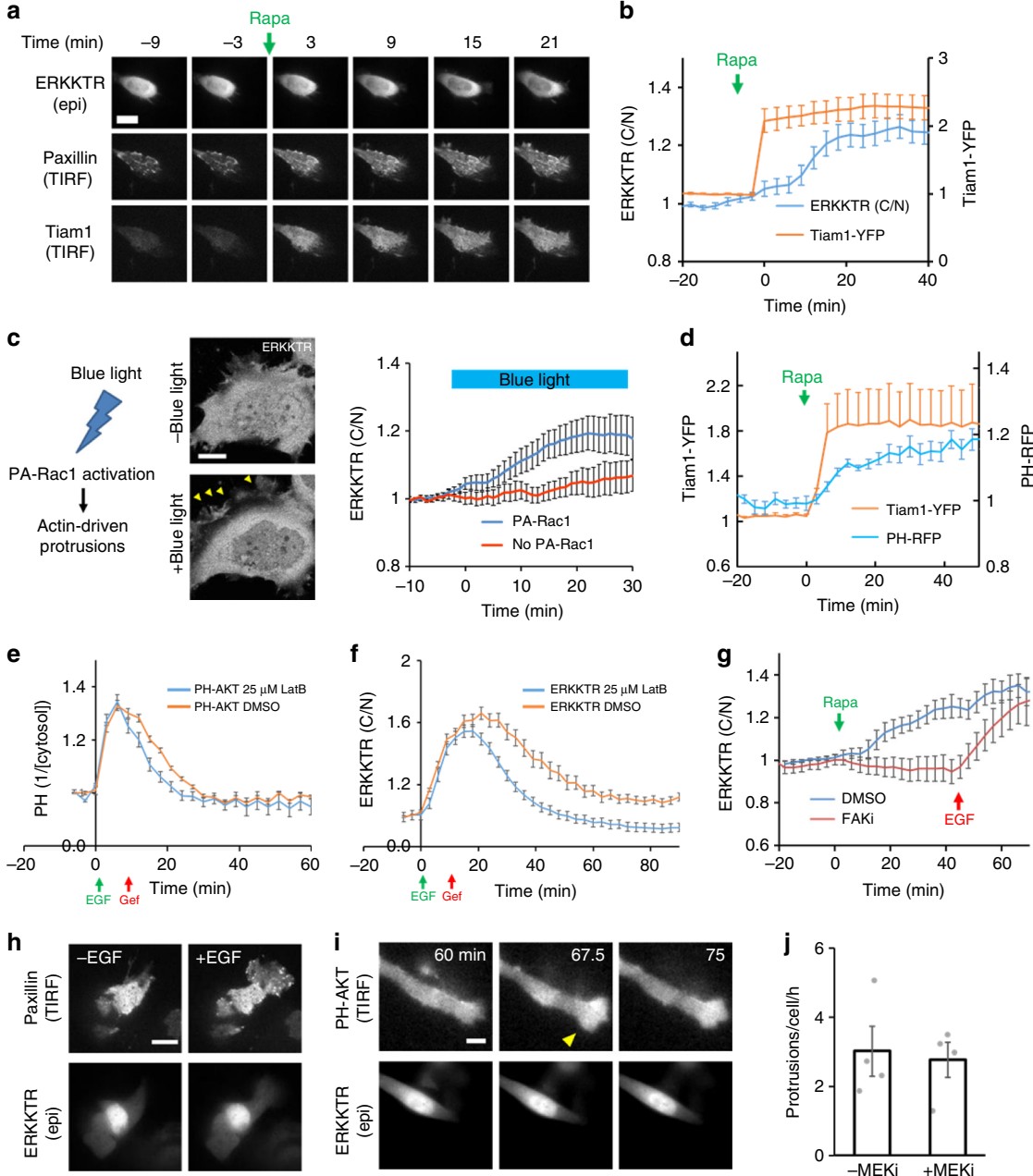

**Fig. 4** ERK activation by artificially induced protrusions. **a**, **b** Representative images for actin polymerization by CID of Tiam1 as well as the corresponding kinetics of YFP-FKBP-Tiam1 and ERKKTR responses (mean ± s.e.m., n = 19 cells). **c** Left: activation of PA-Rac1 by blue light induces actin polymerization and generation of protrusions. Middle: confocal images of a HeLa cell expressing ERKKTR and PA-Rac1 showing generation of protrusions (arrowheads) and nuclear exit of ERKKTR upon exposure to blue light. Right: kinetics of ERKKTR response to blue light in cells with (blue) or without (red) PA-Rac1 (mean ± s. e.m., n = 18 for PA-Rac1 and 16 for no PA-Rac1). **d** Kinetics of CID for Tiam1 and PH-AKT-RFP response (mean ± s.e.m., n = 6). **e**, **f** Responses of PH-AKT (**e**) and ERKKTR (**f**) to transient EGF stimuli (20 ng/ml EGF at 0 min followed by 1 µM EGFR inhibitor gefitinib at 9 min) in the presence of 25 µM Latrunculin B or DMSO (mean ± s.e.m.; normalized to pre-stimulus levels; n = 75 (DMSO) and 20 (LatB) cells for PH-AKT; n = 31 (DMSO) and 57 (LatB) for ERKKTR). **g** Response to CID of Tiam1 in HeLa cells pre-treated with DMSO or 10 µM FAK inhibitor PF-573228 for 1 h (mean ± s.e.m., n = 10 cells each). EGF (100 ng/mL) stimulation was given at the indicated time point (red arrow). See Supplementary Figure 6e for a similar experiment using an alternative FAK inhibitor PF-562271. **h** An MCF7 cell treated with 10 µM MEK inhibitor (MEKi, PD325091) for an hour and stimulated with 20 ng/ml EGF. **i** An SKOV3 cell treated with 1 µM MEKi for a hour showing continued generation of protrusions (arrowheads, see Supplementary Movie 11 for an additional example). **j** Quantification of protrusion frequency by FDM (mean ± s.e.m., n = 4 experiments) in SKOV3 cells before and after treatment with the MEK inhibitor (1 µM PD325901). Scale bars: 10 µm

transient, the excitable activation drove a single peak with a magnitude similar to that of the initial peak induced by a continuous stimulus (Fig. 5e). We tested these predictions by stimulating MCF7 cells with continuous or transient EGF. At the population level, a long stimulus induced a large initial peak of PI3K and ERK activation followed by secondary activities (Fig. 5f). In individual cells, the kymograph of PH-AKT-enriched protrusions was characterized by a global initial response followed by repeated patches at various locations (Fig. 5g, top). Invariably, the protrusions were followed by nuclear exits of ERKKTR, which

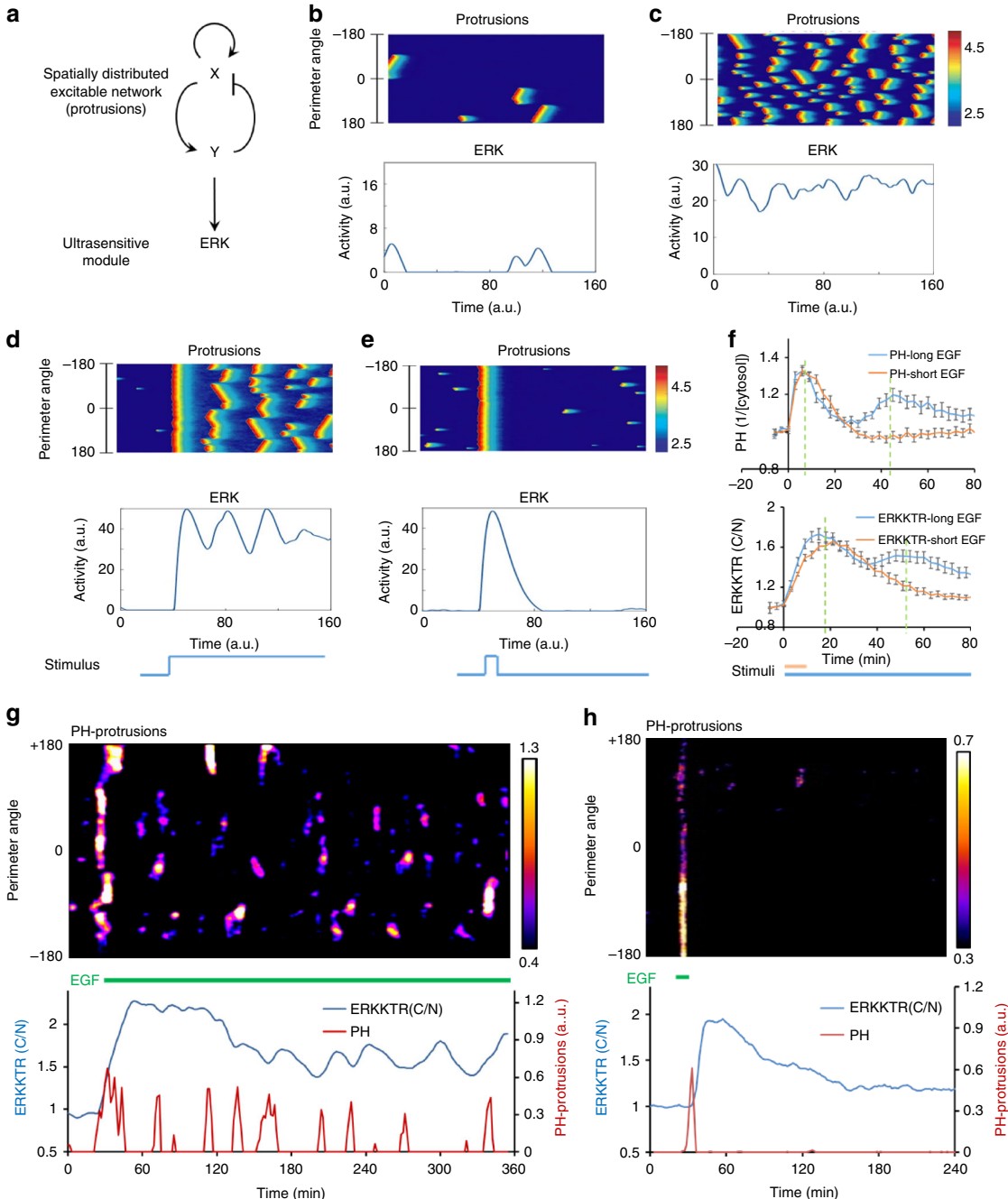

**Fig. 5** Computational simulation and experimental validation of the model. **a** Schematic of a spatially distributed protrusive excitable network (EN) coupled to an ultrasensitive ERK module. **b, c** Simulated kymographs of protrusions along the perimeter of a circular cell (upper) driving ERK activation (lower). A high threshold for EN caused infrequent protrusions with discrete ERK pulses (**b**), whereas a low threshold led to frequent protrusions and continuous ERK activation (**c**). **d, e** Simulated kymograph of protrusions and ERK activation in response to a long (**d**) and short (**e**) stimulus. **f** Responses of PH-AKT (upper) and ERKKTR (lower) to long (20 ng/ml of EGF at 0 min) and transient (20 ng/ml of EGF at 0 min followed by 1 µM gefitinib at 9 min) stimuli in MCF7. Values are mean ± s.e.m., with normalization to pre-stimulus levels; n = 20 (short) and 22 (long) cells for PH-AKT; 31 (short) and 57 (long) cells for ERKKTR. **g, h** (Upper) Kymographs of PH-AKT-enriched protrusions (color scale: intensity (A.U.) identified by FDM, see Methods) around the cell perimeter in response to a long (**g**) and short (**h**) EGF stimulation. (Lower) Plot of integrated intensity of protrusions and C/N ratio of ERKKTR

oscillated with a frequency determined by that of the partially synchronized protrusive activities (Fig. 5g, bottom). In contrast, a short stimulus caused a single peak of PI3K and ERK with its average magnitude and kinetics similar to those of the first peak induced by a long EGF stimulus (Fig. 5f). The kymograph of PH-AKT-enriched protrusions showed a single global response accompanied by one ERK pulse (Fig. 5h). Together the

experimental observations were in excellent agreement with the computational predictions from our model.

**Integration of mechanical and chemical signals.** Why is ERK activation coupled to dynamic protrusions? It is known that ERK signaling responds not only to chemical stimuli such as growth factors, but also to mechanical stimuli, such as matrix

stiffness[39,40]. We speculate that cells can use protrusions to probe the local mechanical properties of their environment. Moreover, the feedback loop between Ras-PI3K, the cytoskeleton, and cellular adhesion may allow various types of stimuli to modulate the excitable network through different entry points. For example, stiff substrates could signal through cellular adhesion, thus lowering the threshold for triggering protrusions by growth factors, which signal through the Ras-PI3K pathway. Thus, different types of stimuli may be integrated by this protrusion-generating excitable network (Fig. 6a). To test this idea, we stimulated cells grown on substrates of different stiffness with escalating EGF doses. The level of phospho-ERK and phospho-FAK increased

with higher EGF concentration, but the sensitivity of their responses to EGF was reduced on soft substrates (Fig. 6b, c). In confocal images, cells on soft substrates appeared round and made fewer protrusions, but ERK pulses were still associated with protrusions regardless of the rigidity of the substrate (Fig. 6d, Supplementary Movie 12). Analysis of ERKKTR nuclear exit events showed that higher EGF caused more frequent ERK pulses; however, at the same EGF concentration, cells on a soft substrate generated significantly fewer ERK pulses (Fig. 6e). Together, the results suggest that cells can integrate chemical and mechanical stimuli by modulating the frequency of protrusion-driven ERK activation.

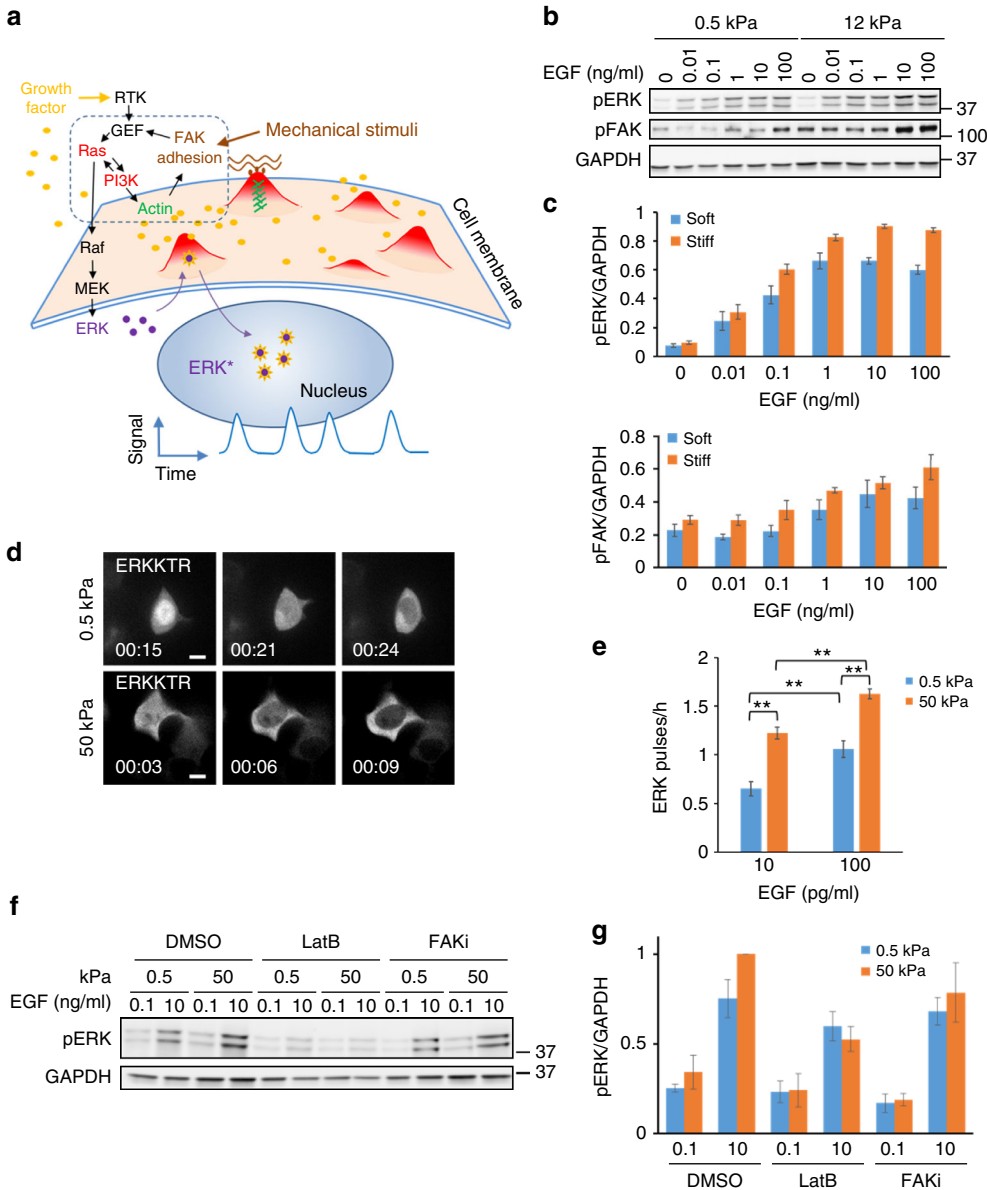

**Fig. 6** Integration of chemical and mechanical stimuli by dynamic protrusions. **a** Model for integration of chemical and mechanical stimuli through dynamic protrusions driven by an excitable network involving a feedback loop between Ras-PI3K, actin cytoskeleton, and adhesion (dashed box). **b** Immunoblots of MCF7 cells on soft and stiff substrates 15 min after stimulation with indicated concentration of EGF. **c** Quantification of **b** (mean ± s.e.m., n = 5 experiments; $p = 6 \times 10^{-9}$ for pERK and 0.001 for pFAK by two-way ANOVA between cells on soft and stiff substrates.) **d** Confocal images of MCF7 cells on 0.5 kPa (top) and 50 kPa (bottom) substrates showing coupling of ERKKTR nuclear exit with protrusions. See also Supplementary Movie 12 for additional examples. Scale bars: 10 μm. **e** Frequency of ERKKTR nuclear exit events in MCF7 cells on stiff and soft substrates incubated with 10 and 100 pg/ml EGF (mean ± s.e.m.; n = 78, 126, 84, 141 from left to right). **p < 0.001 by two-tail unpaired t-test. **f** Immunoblots of MCF7 cells treated with DMSO, 5 μM LatB, or 10 μM FAKi for 1 h prior to EGF stimulation for 15 min. **g** Quantification of pERK levels (mean ± s.e.m., n = 3 experiments, p = 0.049 (DMSO), 0.70 (LatB) and 0.54 (FAKi) by two-way ANOVA between cells on soft and stiff substrates.)

To further test the role of protrusions in chemical and mechanical sensing, we treated cells with latrunculin to abolish protrusion formation. Latrunculin-treated cells still showed increased phospho-ERK with higher EGF concentration, but the response was reduced compared to control cells (Fig. 6f, g). Moreover, the difference of phospho-ERK levels between cells on soft and stiff substrates was eliminated. Consistent with a role for FAK in mediating mechanical sensing, FAK inhibition also abolished the sensitivity to substrate stiffness (Fig. 6f, g). Together these observations demonstrate the importance of protrusions in sensing mechanical stimuli and enhancing responses to chemical stimuli in ERK signaling.

## Discussion

Our study provided important insights into the mechanism and function of dynamic ERK activation at the cellular level. We found that ERK pulses were driven by localized Ras activation on protrusions. Interestingly, internal flashes of Ras did not trigger ERK but obscured the detection of protrusion-associated Ras activities (Supplementary Figure 4), thus explaining the apparent lack of corresponding Ras pulses at the whole-cell level[9]. The stochasticity of ERK activation suggested an underlying excitability of the signaling pathway[9,41]. Excitable phenomena, such as the action potential of neurons, are often characterized by all-or-none responses and refractoriness. ERK activation has indeed been shown to be refractory to repeated growth factor stimuli[42]. However, the sustained activation of ERK in cells with frequent protrusions (Fig. 2g) or stimulated with high concentration of EGF[8] argues against the existence of a true refractory period. Moreover, the magnitude of individual ERK pulses was not all-or-none (Fig. 1d, e). These conflicting observations can be reconciled by noting that the apparent excitability of ERK is derived from an upstream Ras-PI3K signaling network shown to be excitable[14–17,43,44].

Although artificially induced protrusions drove ERK activation (Fig. 3 and 4), acute inhibition of ERK signaling did not affect the generation of protrusions (Fig. 4h–j), suggesting that protrusions themselves, rather than ERK, are the initiator of the observed pulsatile dynamics. However, ERK has also been suggested to regulate cell migration directly through the cytoskeleton and cellular adhesion[45–47], or indirectly through crosstalks with PI3K signaling[48]. Taken together, these observations suggest that ERK is not required to initially generate or cause oscillations of individual protrusions, key behaviors for mechanical environment sensing, but still plays a role in cell motility at a slower time scale.

We also revealed a functional role of protrusions in integrating chemical and mechanical stimuli. At the molecular level, stiff matrices cause clustering of integrins[49], which recruit RasGEFs to the membrane through a pathway involving FAK, Src, Fyn, or Shc[32]. Consistently, we found that FAK is an essential mediator for protrusion-induced ERK activation (Fig. 4g and Supplementary Figure 6e) and sensitivity to substrate stiffness (Fig. 6f, g). However, only newly formed adhesion complexes on protrusions were associated with ERK pulses (Fig. 1i), suggesting that the baseline integrin signaling alone does not efficiently activate Ras, and the amplification by the excitable network is needed. Paxillin in the focal adhesion complex may function as a scaffold protein for the Raf-MEK-ERK cascade[50–52] to further facilitate the activation of the signaling pathway by both chemical and mechanical stimuli.

Our findings also shed new light on the role of Ras signaling in cancer. First, a hallmark of cancer cells is their tendency to metastasize[53]. The ability of protrusions to drive ERK signaling through a feedback between the cytoskeleton and Ras-PI3K signaling indicates that the motility of cancer cells may be coupled to other processes, such as proliferation, metabolism, and protein synthesis. The responsiveness of protrusions to mechanical stimuli may also explain how stiff matrices promote cancer cell invasion and proliferation[54,55]. Second, in previous studies the function of excitability was largely considered in the context of cell migration, in which the stochasticity and amplification associated with Ras-PI3K signaling network play critical roles in random and directed cell movement, respectively[13–20]. Our findings demonstrated the importance of excitability in yet another downstream pathway of Ras GTPases, the MAPK/ERK pathway. We speculate that excitability is a common feature of processes regulated by Ras GTPases. Excitable systems generally involve a positive feedback loop, which initiates a full activation once the threshold is crossed, as well as a negative feedback, which terminate the response and enter the system into a refractory state[56]. Our perturbation analyses indicated that Ras-PI3K, the cytoskeleton, and cellular adhesion mediate the positive feedback process. However, without cytoskeletal activity cells could still generate a large, albeit slightly shorter response to a transient growth factor stimulus (Fig. 4e, f), suggesting the existence of redundant feedback loops. The nature of the negative feedback in the excitable network is unknown, but may involve suppressors of protrusions such as PKB/AKT[57] and membrane tension[58] in migrating cells. Elucidating the molecular mechanism of excitability may reveal strategies for inhibiting Ras activation, which has remained a recalcitrant target for cancer treatment.

## Methods

**Plasmids.** Constructs for Lyn, RBD51-220, PH-AKT, and LifeAct were obtained from Dr. Peter Devreotes' lab. The genes were further cloned into pEGFP, pmCherry, or pcDNA3.1 vectors for tagging with various fluorescent proteins (GFP, mCherry, or CFP). Paxillin, ERKKTR, PA-Rac1, cytoplasmic EKAR constructs were obtained from AddGene (pRK-GFP-Paxillin, #50529; pLentiCMV-Puro-DEST-ERKKTR-Clover, #59150; pTriEx-mCherry-PA-Rac1, #22027; cytoplasmic EKAR (Cerulean-Venus), #18679). The ERKKTR gene was further cloned into pEGFP-N1 and pmCherry-N1 (CloneTech) using primers 5′-CAAgtcga-cATGAAGGGCCGAAAGCCTC-3′ and 5′-CAAggatccccGGATGGGAATT-GAAAGCTGGACT-3′ via SalI/XhoI and BamHI sites. Constructs for chemically induced dimerization (CID) experiments, including Lyn-CFP-FRB, YFP-FKBP-CDC25, YFP-FKBP-iSH2, and YFP-FKBP-Tiam1, were generous gifts from Dr. Takanari Inoue.

**Drugs.** Stocks of 50 mM LY294002 (Cayman, #70920), 5 mM Latrunculin A (AdipoGen, #AGCN20027C100), 25 mM Latrunculin B (Enzo, #BML-T110-0001), 10 mM MEK inhibitor PD325901 (Calbiochem, #444966), 10 mM FAK inhibitor PF-573228 (Cayman, #14924), 10 mM FAK inhibitor PF-562271 (AdipoGen, SYN-1064), 10 mM ZSTK474 (Cell Signaling, #13213), 1 mM Gefitinib (Cayman, #13166), and 10 mM Rapamycin (Cayman, #13346) were prepared by dissolving the chemicals in DMSO. The stocks were diluted to the indicated final concentrations in culture medium. The EGF stock solution was prepared by dissolving EGF (Sigma, #E9644) in 10 mM acetic acid to a final concentration of 1 mg/ml. All drug stocks were stored at $-20\,^{\circ}C$.

**Cell lines.** The cell lines MCF7 (human breast cancer), SKOV3 (human ovarian cancer), 4T1 (mouse breast cancer), HeLa (human cervical cancer), and U2OS (human osteosarcoma) were purchased from ATCC. MCF7, HeLa, and U2OS were grown at 37 °C, 5% $CO_2$, in DMEM high glucose medium (Gibco, #11965092) supplemented with 10% FBS (Corning Cellgro, 35-010-CV), 1 mM sodium pyruvate (Gibco, #11360070), and 1X non-essential amino acids (Gibco, #11140076). 4T1 and SKOV3 were maintained in RPMI 1640 medium (Gibco, #11875093) containing 10% FBS, 1 mM sodium pyruvate, and 1X non-essential amino acids (Gibco, #11140076). Transient transfections of the cancer cells were performed using GenJet In Vitro DNA Transfection Reagent ver. II (SignaGen, #SL100489) following manufacturer's instruction. Cells were transferred to 35 mm glass-bottom dishes (Mattek, Tissue Culture Dish P35GC-0-14-C) and allowed to attach overnight prior to imaging. For substrate stiffness experiments, cells were grown on collagen-coated hydrogels of different stiffness bound to polystyrene plates or dishes purchased from Matrigen (Softwell-6- and Softslip-6-plates for immunoblotting; Softview glass bottom dishes for live cell imaging). Cells were seeded at $4 \times 10^5$ per well or dish and incubated at 37 °C, 5% $CO_2$ overnight before harvest for immunoblotting or live cell imaging.

**Immunoblotting**. Cells were seeded at $4 \times 10^5$ per well in six-well plates with appropriate growth medium and incubated at 37 °C, 5% $CO_2$ overnight, or treated with drugs for indicated period of time before harvesting. Cell lysates were prepared by cell lysis on ice with 1X RIPA buffer (Cell Signaling, #9806) containing protease inhibitor (Roche, #11836170001) and phosphatase inhibitor cocktail (Sigma, #P5726). Immunoblotting of individual protein bands was performed by incubating the PVDF membranes with the following primary antibodies (all purchased from Cell Signaling) diluted 1:1000 in 5% BSA/TBST: rabbit anti-phospho-ERK (#9101), mouse anti-ERK1/2 (#9107), rabbit anti-phospho-AKT (Ser473) (#4060), mouse anti-AKT (#2920), rabbit anti-phospho-FAK (Y397) (#8556), rabbit anti-FAK (#13009), and rabbit anti-GAPDH (#2118). Fluorescent Alexa 488-conjugated donkey anti-rabbit or anti-mouse antibodies (Life Technologies, A21206 or A21202) were used as secondary antibodies diluted 1:5000 in 5% BSA/TBST to visualize the protein bands on the Pharos Molecular Imager (BioRad). Intensities of individual protein bands from the scanned images were measured using ImageJ after subtracting background. Phosphoprotein levels were then normalized to either GAPDH or total level of the corresponding protein. Uncropped images of immunoblots are shown in Supplementary Figures 7 to 11.

**Live cell imaging**. To detect Ras activation, we used fluorescent protein (FP)-tagged RBD (residues 51–220 of Raf-1 containing a Ras-binding domain plus a cysteine-rich domain) that binds to the activated form of endogenous Ras on the membrane[59]. To detect PI3K activation, we used FP-tagged PH-AKT (PH domain from AKT), which binds to phosphatidylinositol (3,4,5)-trisphosphate (PIP3) produced by PI3K. Wide-field epifluorescence microscopy revealed that activation of Ras and PI3K by EGF stimulation led to robust membrane recruitment of RBD and PH-AKT (Supplementary Figure 1a, b). To visualize the activities on the membrane, we imaged RBD-GFP and PH-AKT-RFP along with a membrane marker Lyn-CFP in different cell lines using TIRF microscopy. The intensity of RBD and PH-AKT fluorescence was normalized to that of Lyn.

Wide-field epifluorescence, TIRF, and confocal microscopy have been described previously[15]. Briefly, epifluorescence and TIRF microscopy experiments were carried out on a Nikon Eclipse TiE microscope illuminated by an argon laser (GFP) and a diode laser (RFP). Images were acquired by a Photometrics Evolve electron multiplying charge-coupled device camera (EMCCD) camera controlled by Nikon NIS-Elements. Confocal microscopy was carried out on either Zeiss LSM780 single-point laser-scanning microscope (Zeiss AxioObserver with 780-Quasar confocal module; 34-channel spectral, high-sensitivity gallium arsenide phosphide detectors, GaAsP) controlled by the Zen software, or 3i Marianis/Yokogawa Spinning Disk Confocal microscope controlled by the 3i Slidebook software. All live cell imaging were carried out in a temperature/humidity/$CO_2$-regulated chamber. For high-speed volumetric data acquisition with low phototoxicity, we used a custom lattice light-sheet (LLSM) microscope built by Intelligent Imaging Innovations (3i) based on the designs from Eric Betzig[27]. Briefly, an AOTF-controlled, collimated laser beam is passed through a cylindrical lens telescope to form a stripe. This stripe is patterned using a ferroelectric spatial light modulator (Forth Dimension Displays), then passed through a Fourier transform lens and further refined at the back focal plane using an annular mask. A pair of galvanometers (termed the "X galvo" and "Z galvo"), also conjugate to the back focal plane, allow this pattern to be quickly moved in X and Z at the imaging plane. Finally, this pattern is sent through a custom water-dipping excitation objective (Special Optics, 0.65NA). This serves to generate a very thin array of parallel pencil-like beams, which are dithered using the X galvo to generate a light sheet. A second emission objective (Nikon CFI Apo LWD 25X/1.1) mounted at 90 degrees to the excitation objective serves to collect light from the excitation plane, which passes through a laser blocking filter and is imaged on a Hamamatsu Orca Flash V2 + sCMOS camera. The sample is mounted on a 5 mm round coverslip, which is immersed in a heated media bath and held between the two objectives by a lightweight cantilever. The cantilever is mounted to a motorized stage to allow for 3D translation, as well as a piezo translator for fast, highly reproducible movement of the sample through the light sheet during imaging. By using an FPGA to synchronize laser pulses, SLM pattern presentation, and camera exposure with piezo translation, near-confocal resolution image stacks are obtained at the speed of the camera. As the piezo is moving at an angle to the imaging plane, the resulting data is skewed. Deskewing, as well as deconvolution (to improve image quality) and cropping (to reduce data size) is performed by custom software provided by Eric Betzig. Videos of LLSM images were generated with the Imaris software.

**Chemically induced dimerization (CID)**. To control target protein activation temporally and spatially through CID, cells were transiently transfected with YFP-FKBP-tagged target protein (Cdc25, iSH2, or Tiam1) along with the membrane tethered Lyn-CFP-FRB. The culture medium was replaced with Leibovitz's L-15 supplemented with 1% FBS 18 h after transfection, and transfected cells were incubated in this low-serum medium overnight before imaging. To activate target proteins, 5 μM rapamycin was added to the cells to induce FRB-FKBP dimerization, thus recruiting target protein to the plasma membrane. Cellular responses were captured by the Nikon Eclipse Ti-E microscope equipped with a Photometrics Evolve 512 EMCCD.

**Optogenetic induction of protrusions by PA-Rac1**. To detect ERK activation upon protrusion generation by photoactivation, cells were transfected with ERKKTR-GFP along with mCherry-PA-Rac1[30]. Cells were starved in Leibovitz's L-15 medium containing 1% FBS overnight prior to imaging. Time-lapse videos were captured on either a Nikon Eclipse Ti-E or Zeiss LSM780 microscope. Protrusions were induced with repeated blue light irradiation. For Nikon Eclipse Ti-E, the whole viewfield under 100X objective was irradiated with 440 nm laser at 3% power for 5 s every 20 s. For Zeiss LSM780, the whole viewfield under 63X objective was irradiated with 458 nm laser at 40% power for 10 iterations (~25 s) every 1 min.

**Image processing and analysis**. Microscopy images were processed and analyzed with NIH ImageJ and Fiji[60,61]. For cytoplasmic:nuclear (C/N) ratio calculation of ERKKTR, the nuclear region can be determined based on the shuttling of ERKKTR with or without a second marker such as paxillin, which shows a diffuse nucleus excluded, cytoplasmic fluorescence. Frame Difference Method (FDM) was used to analyze temporal changes of biosensor activities in time-lapse images. The method was based on calculating the pixel-by-pixel difference in intensity between frames separated by defined intervals in the time-lapse video. To minimize the effect of noise, 2–4 consecutive frames were averaged for the calculation. The percentage of change was then obtained by normalizing the difference to the average intensity. By setting suitable thresholds and ranges for the percentage change rate, FDM was used to analyze the frequency of protrusions (Figs. 2d–f, 3d, 4j, Supplementary Figure 4c), generate the kymograph of protrusions enriched for biosensors (Figs. 1d, 2g, 5g, h), and display the internal flashes of Ras-PI3K activities (Supplementary Figure 3a-c, 4c, d). The thresholds for different types of activities were determined empirically based on visual comparison of original and FDM-processed images. For analysis of protrusions, an empirical threshold of 100% intensity change (i.e. a twofold increase) over 6 min was used. The quantification of protrusion frequency was automated using the object tracking plugin MTrack2, by setting the minimum duration (5 min) and pixel size of the protrusions. For kymograph generation, a segmented line was created around the perimeter of the cell in FDM-processed images. The KymographBuilder plugin was then used to generate a kymograph along the line. For internal flashes of Ras-PI3K activities, the threshold was set at 10% over 6 min after removal of boundary pixels using a mask generated by the Binary tool in ImageJ.

FRET experiments were carried out as described previously[62]. Briefly, cells expressing EKAR were imaged on a Nikon Eclipse TiE microscope illuminated by a diode laser (440 nm) with an Evolve EMCCD controlled by Nikon NIS-Elements. A Dual-View system (Optical Insights, LLC) was used for simultaneous imaging of cyan fluorescent protein (CFP) and yellow fluorescent protein (YFP) fluorescence in cells expressing EKAR Cerulean-Venus. The ratio of YFP to CFP intensity was used as proxy for FRET efficiency.

For cross-correlation analysis, correlation coefficients were calculated between ERK activity (measured by ERKKTR C/N or EKAR FRET) and the area of protrusion with the specified time lag. The cross-correlation coefficient is then graphed in the y-axis against the time lag in the x-axis.

**Statistics**. At least two independent experiments were carried out on different days for imaging studies. At least three independent experiments were carried out on different days for immunoblotting. Statistical significance and p-values were determined using the two-tail unpaired Student's t-test for comparison between two groups. Two-way ANOVA was used for comparing the responses of phospho-ERK and phospho-FAK to different doses of EGF for cells on stiff and soft substrates (Fig. 6c–g). Mean ± s.e.m. (standard error of the mean) was reported as indicated in the figure legends. Statistics were derived by aggregating the n number of samples noted in each figure legend across independent experiments.

**Computational modeling**. The excitable Ras-PI3K network is modeled as a two-species activator-inhibitor system. The activator (X) is autocatalytic, i.e. it stimulates its own production once the threshold for activation is crossed. The activator simultaneously initiates a negative feedback loop through the inhibitor (Y) that slowly subdues the activator response. The system can be described by the following partial differential equations[63]:

$$\frac{\partial X}{\partial t} = D_X \nabla^2 X - a_1 X - a_2 (Y - R) X + \frac{a_3 X^2}{a_4^2 + X^2} + a_5 + U_N. \tag{1}$$

$$\frac{\partial Y}{\partial t} = D_Y \nabla^2 Y + \varepsilon(-Y + b_1 X). \tag{2}$$

Both the components in this system can diffuse spatially, with diffusion coefficients $D_X$ and $D_Y$, respectively. The signals $R$ and $U_N$ are inputs to the excitable system. They refer to an external stimulus (through EGF) or a stochastic input, respectively. The stochastic component is modeled as a zero mean, white noise process with a constant variance. The parameters $b_1$ and the noise variance were modulated to simulate the low and high activity of the excitable system. The ERK module is modeled as a zero-order ultrasensitive switch (state Z) that is coupled to the excitable system described above. The model is setup through an

ordinary differential equation as shown below[64]:

$$\frac{dZ}{dt} = c_1 \left( \frac{U_E(c_2 - Z)}{c_3 + (c_2 - Z)} - \frac{c_4 Z}{c_3 + Z} - c_5 Z \right). \qquad (3)$$

The third term is added to the ultrasensitive switch equation to enhance the decay rate of the state. As mentioned in the text, the ultrasensitive module is global in nature and thus diffusion of the state is ignored. The input to the ERK module is $U_E$, which is modeled as the mean output of the $Y$ state from the excitable system.

Parameters used in these simulations are provided in Supplementary Table 1. The model and all simulations are implemented using MATLAB. The PDEs for the excitable system were solved by representing the cell boundary as a one-dimensional system—discretized in space using 300 points. Spatial diffusion terms, which contain the second derivatives, were approximated by central differences in space, subsequently converting the partial differential equations to ordinary differential equations. The time step for simulation was set to 0.01 s. The solutions of the stochastic differential equations were obtained using the SDE toolbox for MATLAB[65].

**Code availability**. The model and all simulations are implemented using Matlab (MathWorks). The computer code is available from the corresponding author upon request.

## Data availability
All data supporting the findings of the current study are available within the article and its Supplementary Information files or from the corresponding author upon reasonable request.

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

## Acknowledgements
The authors would like to thank Takanari Inoue, Peter Devreotes, Douglas Robinson, Sergi Regot, and Feng-Chiao Tsai for helpful discussions; the Inoue lab for constructs for chemically induced dimerization; and Jennifer Chang and Liangmei He for technical assistance. This work was supported in part by grants from DARPA HR0011-16-C-0139 (to P.A.I.) and the National Institutes of Health P50CA098252 (to T.C.W.), R01CA183040 (to T.C.W.), K22CA212060 (to C.H.H.), and a Cervical Cancer SPORE Career Development Award (to C.H.H.). The LLSM was purchased with NIH Grant S10 OD018118.

## Author contributions
J-.M.Y. and C.-H.H. conceived the project, designed the experiments, collected, analyzed the data and wrote the manuscript. S.B. and P.A.I. performed computer simulations. H.W.-F. provided technical support for lattice light-sheet microscopy. J-.M.Y., S.B., H.W.-F., C.-F.H., T.-C.W., P.A.I., and C.-H.H. participated in the preparation of the manuscript. C.-H.H. supervised the study.

## Additional information

**Competing interests:** The authors declare no competing interests.

