## [Peer Review File · Nature Communications]

Reviewers' comments:

Reviewer #1 (Remarks to the Author):

This manuscript by Yang et al. makes the interesting link between protrusion activity (a mechanical event) and ERK signaling (a chemical event). If this is true, then this work will add to a rapidly increasing body of literature that underlines the importance of mechanical events in signaling homeostasis, far beyond the well-established feedbacks between forces and signals implicated in cell morphogenesis per se. The authors do not discuss this too prominently (which I applaud as a good exception to the notorious over-hype of many papers), but this link between protrusion and ERK activity could be of central importance in cancer. Experimentally, the paper uses a series of translocation sensors for Ras, PI3K, and ERK to measure signaling activity, along with Western blots and pharmacological inhibition studies of cells at the population level. The most innovative aspect is the use of the CID (chemically induced dimerization) approach to activate signaling components. The authors also build a very simple mathematical model to illustrate the effect of excitable protrusions on the integration of protrusion pulses by ERK. There is nothing wrong with the model, but given the set of underlying assumptions the insights the model offers are not particularly surprising. I consider this model more as a replacement of the standard final cartoon rather than a true driver of the investigation.

My main two concerns with the current status of the study are insufficient rigor of the data and absence of a convincing mechanism that links actin-mediated cell protrusion to ERK activation. For the rigor, overall the image and movie quality is not up to par with comparable studies in the literature. Most importantly, the evidence that protrusion may precede ERK activation is anecdotal at best (and from this the authors immediately jump to the conclusion that protrusion is causative, although the definition of causation is complex in a system with as many feedbacks as the one studied here). The authors provide one example in Fig. 1f. There is no attempt to establish statistical significance of this relation across biological repeats. I gather from the text that not all protrusions relate to ERK activation. No algorithm is proposed for filtering this data in an unbiased fashion. The authors do not provide a key control for the ERK activity measurement, which relies on the cytoplasm to nuclear (C/N) intensity ratio of a probe that shuttles between the two compartments. The images are recorded by epifluorescence, which is highly susceptible to cell volume changes. For example, as the cell protrudes, the cell spreads and dependent on the focus setting more of the probe may contribute to the signal, resulting in an increased C/N ratio. Hence, the positive correlation of protrusion and ERK activation may result, in part, from an imaging artifact. While some of this probably could be accounted for by separation of time scales the simplest approach will be normalization relative to a volume marker. I feel, this is the minimal control experiment the authors should add. A similar normalization issue applies to the WBs of ERK activity. It is unusual to normalize phospho-ERK to the loading control rather than the kinase itself (Fig. 2). Also, why does the WB in Fig. 2b show two bands for pERK?

For the mechanism, unfortunately the paper leaves the reader really in the dark as to how ERK senses the protrusion activity. On the sidelines of Fig. 3a the authors suggest that there might be a feedback from actin assembly to adhesion to Ras signaling, which then activates the MAPK cascade to a higher level. The authors mention then also a few other possible pathways, but none of the proposals is systematically tested. In my

opinion, the observation of a potential coupling of cell protrusion to ERK activation is of such high significance to cell biology that this story should be made more complete, if the controls really hold up.

Reviewer #2 (Remarks to the Author):

In the manuscript entitled "Integrating chemical and mechanical signals through dynamic coupling between protrusions and ERK activation" the authors explore the connection between cellular protrusions and ERK activity. Using paired co-expression of fluorescent reporters for ERK, PI3K, and Ras activities, the authors found that protrusions containing active Ras and PI3K occur prior to ERK activation. Cross-correlation of the kymographs reveals a high correlation between peaks of active-Ras protrusions and ERK activity spikes. The authors validate the ERK sensor across several cell lines by comparing the cytoplasm/nuclear ratio of the sensor to western blots for phospho-ERK and show the reporter seems to be broadly representative of active ERK. Using and rapamycin-induced recruitment of activators of Ras, PI3K, or Rac1 the authors show ERK activity correlates with recruitment of these activators to the membrane. The authors then develop a mathematical model where dynamic protrusions stimulate ultrasensitive ERK activity. The model predicts that a single broad protrusion event would generate a large single spike of ERK activity, but smaller stochastic protrusions would generate multiple peaks of ERK activity. Indeed, when the authors stimulate serum-starved with a short pulse of EGF or a continued dose of EGF they see a large single spike or multiple spikes of ERK activity respectively. Finally, the authors determine the dose response of ERK activity to EGF stimulation is similar in conditions of low and high matrix stiffness, but find a higher peak response under high stiffness.

The main conclusion of the paper that protrusions drive ERK pulses is based almost entirely on correlative data and no mechanistic links are established. Since the data is largely correlative, further more direct experiments are necessary to draw such a conclusion. The core experiment using rapamycin to induce protrusive activities and measure ERK response are fairly indirect and have several issues as explained below that raise larger concerns about the conclusions the authors make. Additionally, because previous studies have reported that ERK activity drives protrusion and migration rather than their hypothesis (Aoki et al 2017, Mendoza et al 2011, Chernyavsky et al 2005, and others), it is necessary for the authors to go beyond these studies and directly confirm their model that protrusion is not preceding but instead drives ERK activity. They need to put their work into the body of work from other labs that it is instead ERK activity that drives protrusion. Further, the ERK KTR sensor the authors cannot measure localized ERK activity within the cell (only global activity). An ERK FRET sensor (such as one used in Aoki et al 2017) would provide critical spatiotemporal information on where ERK is activated relative to the Ras/PI3K activity in protrusions. Since the manuscript is correlative and does not provide significant novel mechanistic insights, it is not suitable for publication in Nature Communications.

Specific Points:

1) In Figure 3 using the rapamycin-induced recruitment of Cdc25, iSH2, and Tiam1 the ERK activity is already increasing prior to the addition of rapamycin as shown in graphs in panels d, f, and h. This is very concerning and since these are the only experiment

artificially inducing different protrusive activities and measuring the ERK response it doesn't strongly show what the authors claim. Did all individual cells show ERK increases prior to rapamycin, or is the heterogeneity of the 8 cells averaged creating this? Since this experiment is critical to the authors main conclusion they must show this more convincingly.

2) In Figure 1f it is unclear why some Ras- peaks are called and others not, and also how ERKKTR peaks are defined since they vary in slope and size. The authors should more carefully explain this in the text. Also these plots allow for a more detailed analysis that was not sufficiently explored. What is the typical delay between a protrusive peak and an ERK spike? What are the distributions of delay times and can you predict the size or delay time of the ERKKTR peak from the size of the protrusive Ras peak?

3) Figure 1f and 1g, 4g and 4h, so show the relationship between ERKKTR and protrusions in just one cell. Since the conclusions of the paper rely on mostly correlative data like this, the authors should expand their analysis to multiple cells.

4) In Figure 2d it is unclear how this plot is generated. It is a single cell, what is the scale, should the axis have labels?

5) What is the temporal limitation of the ERKKTR sensor? Have the authors compared the kinetics to available ERK FRET sensors? Since the authors conclude protrusions stimulate ERK activity it would be more convincing if they used a sensor with rapid representation of ERK activity. Experiments comparing dynamics of the ERKKTR to the FRET reporter would be very useful to address this.

6) In figures 1b, 1c, 2g, 4b, 4c, 4d, 4e, 4g, 4h and there is no colormap to indicate the enrichment levels of the sensors shown

7) The stiffness of matrix used is vastly different in the western blot experiments and the ERKKTR experiments shown in Figure 5 panels b, c, d, and e (12 vs 50 kPa). Does the KTR sensor not show difference in the conditions used in the western blot experiment? If not, what does the western blot of pERK show under these conditions?

8) In figure 4h the imaging time for a short pulse of EGF is significantly shorter than the continuous EGF exposure shown in 4g. The authors should ensure there are not other delayed ERKKTR activities or protrusions they are missing.

Reviewer #3 (Remarks to the Author):

Yang et al. perform an analysis of live-cell reporter data to demonstrate a mechanistic linkage between Ras-mediated cytoskeletal activity and overall cellular ERK signaling patterns. Sporadic pulses of ERK activity have been observed in a number of studies, both in vivo and in vitro, but the origin of these kinetics is not well understood. The authors demonstrate that bursts of ERK activity coincide in time with cellular protrusions that are high in Ras activity, as judged by recruitment of Raf-RBD to the plasma membrane. They show that overall protrusive activity predicts ERK signaling activity across multiple cell lines, and also demonstrate that induction of actin signaling by engineered signaling molecules can evoke ERK activation. They develop a model of this connection as an excitable network, with a positive feedback at the level of actin polymerization driving a highly cooperative ERK response.

Overall this is an interesting development in understanding what shapes ERK activity kinetics within the cell, and these observations will be of interest to other researchers focusing on the systems biology of Ras and MAPK regulation. However, there are some

important limitations of this work that make it difficult to recommend for publication in its current form:

1. The quantitative and statistical analyses of the phenomena presented are very limited. For example, a cross-correlation analysis would be an excellent way to characterize the data shown in figures 1F or 4G. It would also be very helpful to see some indication of the trends shown in larger sample of cells, rather than just a single example in each case. Without more extensive analyses it is very difficult to judge the strength of these associations, and therefore to assess the importance of the mechanisms reported.

2. Correlations between signals (For example Figure 1F/G) are reported as relationships between peaks. Conceivably, peaks may not be the most important feature of the relationship between the signals. Comparisons of onset time, or rate of change, would also be very informative, especially given the importance of ordering of events for the model. Plots of the derivatives of the signals would be very helpful. Also, it is not always clear in which cases the frame difference method is being applied.

3. The authors make a distinction between peripheral protrusion-related RBD/PH-Akt pulses that drive ERK activity and interior flashes that do not. While there is some discussion in Fig. S3 of how these differ, it is still not clear how this distinction is drawn from the data. A more comprehensive reporting of the data from multiple cells, as discussed above, would help a lot.

4. The mechanism underlying the kinetics shown remains somewhat obscure. The model described in Fig 4, for example, does little to suggest how specific molecular events create the positive feedback loops that are apparent in the data. One thing that is not yet clear is whether ERK is part of the positive feedback loop, although it is assumed to be downstream. The data from the PI3K and MEK inhibitor experiments are very confusing as presented. The PI3K inhibitor appears to have very different effects on ERK depending on the cell line used, and its effect on protrusions is hard to assess from the pictures shown in Fig. S4D - why not report protrusions/time as in Fig. S4C? The effects of the MEK inhibitor on protrusion are also unclear - only a single cell is shown, in movie form; quantitation would be more helpful. Overall, the data presentation just seems sloppy on this point, and from the current presentation it is hard to see how strong conclusions can be drawn regarding the feedback loops.

Response to reviewer comments

We thank the reviewers for their helpful comments and suggestions. We carried out additional experiments and analyses that not only strengthened our conclusions but also provided important new insights. To accommodate the new information, we have added new figures (Figs. 1f, 1g, 1h, 4c, 4g, 4h, 4i, 4j, 6f, 6g, Supplementary Figs. 2a-e and 4e), updated original figures (Figs. 1a, 2d, 3a, 4b, 5h, 6a, and Supplementary Figs. 4b, 4c, 4d, 5b), and made extensive changes to the text as can be seen from the attached “Comparison between the original and new versions”. The most important new additions are highlighted in the revised manuscript and summarized as follows:

1. **Biosensor validation:** We used a volume marker to rule out the effect of cell morphological changes on cytoplasm-to-nucleus (C/N) ratio of ERKKTR in reporting ERK activity (Supplementary Fig. 2a), and applied an additional ERK reporter, the FRET-based EKAR, to confirm our observations regarding the coupling between protrusions and ERK activation (Fig. 1g, 1h, Supplementary Fig. 2c-e).
2. **The timing between protrusions and ERK activation:** We used two different methods to determine the lag time between protrusions and ERKKTR (C/N): (1) cross-correlation between the protrusion area and ERKKTR (C/N) revealed a lag of 6 min (Fig. 1f); (2) the time to reach half-maximal peak for ERKKTR (C/N) was 5.02 ± 0.75 min (mean \pm S.E.M) behind that of protrusions (Supplementary Fig. 2d). We carried out similar analyses with the FRET-based ERK reporter EKAR. Unexpectedly, we found no significant delay between EKAR signal increase and protrusions (Fig. 1h, Supplementary Fig. 2d). The discrepancy between ERKKTR and EKAR was consistent with their activation kinetics upon EGF stimulation (Supplementary Fig. 2e).
3. **Mechanisms of the coupling between protrusions and ERK activation:**
 - a. In light of the lack of temporal separation between ERK and protrusion activities demonstrated by our EKAR experiments, the relationship between these events had to be determined by perturbational studies. In the original manuscript, we reported that artificially induced protrusions through CID of Ras, PI3K, and Tiam1 led to ERK activation. In the revision we strengthened the conclusion by showing that ERK can also be activated by optogenetically induced protrusions using PA-Rac1 (Fig. 4c).
 - b. We carried out additional experiments and quantifications to demonstrate that inhibition of ERK did not affect EGF-stimulated (Fig. 4h) or spontaneous (Fig. 4i, j) protrusions. Together with the activation of ERK by artificially triggered protrusions by chemical and optogenetic approaches, our results indicate that protrusions are the driver for the pulsatile ERK activation.
 - c. We showed that inhibition of actin by latrunculin abolished phospho-FAK (Supplementary Fig. 6b). FAK inhibition caused dose-dependent reduction in pERK and pAKT (Supplementary Fig. 6c, d) and blocked ERK activation

induced by CID of Tiam1 (Fig. 4g). These observations indicate that FAK mediates the activation of Ras-ERK pathway by actin-driven protrusions (Fig. 3a and Fig. 6a).

- d. By studying the effects of latrunculin and FAK inhibition on the EGF responses of cells on substrate of different stiffness, we demonstrated that protrusions enhance cellular responses to EGF and mediate mechanosensing through FAK (Fig. 6f, g).

Point-by-point response

Reviewer #1 (Remarks to the Author):

This manuscript by Yang et al. makes the interesting link between protrusion activity (a mechanical event) and ERK signaling (a chemical event). If this is true, then this work will add to a rapidly increasing body of literature that underlines the importance of mechanical events in signaling homeostasis, far beyond the well-established feedbacks between forces and signals implicated in cell morphogenesis per se. The authors do not discuss this too prominently (which I applaud as a good exception to the notorious over-hype of many papers), but this link between protrusion and ERK activity could be of central importance in cancer. Experimentally, the paper uses a series of translocation sensors for Ras, PI3K, and ERK to measure signaling activity, along with Western blots and pharmacological inhibition studies of cells at the population level. The most innovative aspect is the use of the CID (chemically induced dimerization) approach to activate signaling components. The authors also build a very simple mathematical model to illustrate the effect of excitable protrusions on the integration of protrusion pulses by ERK. There is nothing wrong with the model, but given the set of underlying assumptions the insights the model offers are not particularly surprising. I consider this model more as a replacement of the standard final cartoon rather than a true driver of the investigation.

Authors' reply: We thank the reviewer for pointing out the importance of our findings in cancer. We have expanded the discussion on the implications for cancer (see Discussion). The reviewer's point regarding the mathematical model is well taken, and we agree that the simulated activities in unstimulated cells were expected. However, we believe that the simulated responses to growth factor stimuli provided important new insight into the basis of ERK oscillation: although localized protrusions following growth factor stimulation has been reported in the past (Aoki 2004, 2005), to our knowledge the link between synchronization of local protrusions and oscillatory ERK activation has never been proposed.

My main two concerns with the current status of the study are insufficient rigor of the data and absence of a convincing mechanism that links actin-mediated cell protrusion to ERK activation. For the rigor, overall the image and movie quality is not up to par with comparable studies in the literature. Most importantly, the evidence that protrusion may precede ERK activation is anecdotal at best (and from this the authors immediately jump to the conclusion that protrusion is causative, although the definition of causation is complex in a system with as many feedbacks as the one studied here).

Authors' reply: As mentioned above, we confirmed that ERKKTR pulses lagged behind protrusions through quantitative analysis across cells. However, our new analysis based on the FRET biosensor EKAR did not show statistically significant time differences between protrusions and ERK activation (Fig. 1h, Supplementary Fig. 2d). Instead, the causative relationship between these events was established through the following observations: (1) artificially induced protrusions, by CID (Fig. 3, 4) and our new optogenetic experiments using PA-Rac1 (Fig. 4c), triggered ERK activation; (2) inhibition of ERK did not block stimulated or spontaneously generated protrusions (Fig. 4h, i, j). These results indicate that protrusions initiate the pulsatile ERK dynamics.

The authors provide one example in Fig. 1f. There is no attempt to establish statistical significance of this relation across biological repeats. I gather from the text that not all protrusions relate to ERK activation. No algorithm is proposed for filtering this data in an unbiased fashion.

Authors' reply: We have expanded the analysis to ERK activation events from multiple cells (Fig. 1f, Supplementary Fig. 2d). We included all large, discrete ERK pulses responses, defined as over 20% change in ERKKTR C/N. In these cases the reversal of nuclear and cytoplasmic intensity of ERKKTR was evident, and the associated protrusions could be unambiguously identified. Among 12 such ERK pulses, 11 were preceded by discrete protrusions, whereas one showed diffuse but small cellular spreading. As mentioned above, we used two different methods to determine the lag time between protrusions and ERKKTR (C/N): (1) cross-correlation between the protrusion area and ERKKTR (C/N) revealed a lag of 6 min (Fig. 1f); (2) the time to reach half-maximal peak for ERKKTR (C/N) was 5.02 ± 0.75 min (mean \pm S.E.M) behind that of protrusions (Supplementary Fig. 2d). We carried out similar analyses with the FRET-based ERK reporter EKAR. Unexpectedly, we found no significant delay between EKAR signal increase and protrusions (Fig. 1h, Supplementary Fig. 2d). The discrepancy between ERKKTR and EKAR was consistent with their activation kinetics upon EGF stimulation (Supplementary Fig. 2e).

The authors do not provide a key control for the ERK activity measurement, which relies on the cytoplasm to nuclear (C/N) intensity ratio of a probe that shuttles between the two compartments. The images are recorded by epifluorescence, which is highly susceptible to cell volume changes. For example, as the cell protrudes, the cell spreads and dependent on the focus setting more of the probe may contribute to the signal, resulting in an increased C/N ratio. Hence, the positive correlation of protrusion and ERK activation may result, in part, from an imaging artifact. While some of this probably could be accounted for by separation of time scales the simplest approach will be normalization relative to a volume marker. I feel, this is the minimal control experiment the authors should add.

Authors' reply: Following the reviewer's suggestion we validated the nucleocytoplasmic shuttling of ERKKTR by normalization to the volume marker GFP (Supplementary Fig. 2a). Moreover, we used an alternative ERK reporter, the FRET-based EKAR, to confirm the association between protrusions and ERK activation (Fig. 1g, h, Supplementary Fig. 2c, d).

A similar normalization issue applies to the WBs of ERK activity. It is unusual to normalize phospho-ERK to the loading control rather than the kinase itself (Fig. 2). Also, why does the WB in Fig. 2b show two bands for pERK?

Authors' reply: For most immunoblotting experiments involving drug treatment (e.g. Supplementary Fig. 5 and 6), control and treated cells were derived from a homogeneous source and the loading control (i.e. GAPDH) should reflect the total kinase amount for the purpose of normalization. Not all antibodies against the total proteins (e.g. AKT, ERK, or FAK) worked equally well especially for low abundant species. Therefore, for the sake of consistency we opted to normalize the phospho-proteins against GAPDH. Nevertheless, the reviewer rightly pointed out that the situation is different for Fig. 2b, which compares different types of cells. In this case, pERK/ERK and pERK/GAPDH reflect the percentage and total amount of active ERK in each cell, respectively. ERK1 is an ERK substrate that changes its C/N distribution depending on the total activity of ERK rather than the percentage of active ERK in the cells. Therefore we compared ERK1 C/N ratio to pERK/GAPDH instead of pERK/ERK. The two bands of pERK corresponded to phosphorylated ERK1 and ERK2 (p42 and p44), which are not distinguishable by most commercial ERK antibodies. We include the information in the updated figure legend.

For the mechanism, unfortunately the paper leaves the reader really in the dark as to how ERK senses the protrusion activity. On the sidelines of Fig. 3a the authors suggest that there might be a feedback from actin assembly to adhesion to Ras signaling, which then activates the MAPK cascade to a higher level. The authors mention then also a few other possible pathways, but none of the proposals is systematically tested. In my opinion, the observation of a potential coupling of cell protrusion to ERK activation is of such high significance to cell biology that this story should be made more complete, if the controls really hold up.

Authors' reply: We conducted additional experiments to shed light on the mechanism of protrusion-induced ERK activation. First, inhibition of actin by latrunculin abolished phospho-FAK as well as paxillin patches (Supplementary Fig. 6a, b). Second, FAK inhibition caused dose-dependent reduction in pERK and pAKT (Supplementary Fig. 6c, d). Third, FAK inhibition blocked ERK activation induced by CID of Tiam1 (Fig. 4g). Together, these observations suggest that actin-driven protrusions activate FAK, which mediates the activation of Ras-ERK pathway. FAK has been shown to generate phosphotyrosine sites that recruit RasGEFs, leading to Ras activation (e.g. reviewed in Giancotti 1999). We included this pathway in the updated Fig. 3a and Fig. 6a.

Reviewer #2 (Remarks to the Author):

In the manuscript entitled "Integrating chemical and mechanical signals through dynamic coupling between protrusions and ERK activation" the authors explore the connection between cellular protrusions and ERK activity. Using paired co-expression of fluorescent reporters for ERK, PI3K, and Ras activities, the authors found that protrusions containing active Ras and PI3K occur prior to ERK activation. Cross-correlation of the kymographs reveals a high correlation between peaks of active-Ras protrusions and ERK activity spikes. The authors

validate the ERK sensor across several cell lines by comparing the cytoplasm/nuclear ratio of the sensor to western blots for phospho-ERK and show the reporter seems to be broadly representative of active ERK. Using and rapamycin-induced recruitment of activators of Ras, PI3K, or Rac1 the authors show ERK activity correlates with recruitment of these activators to the membrane. The authors then develop a mathematical model where dynamic protrusions stimulate ultrasensitive ERK activity. The model predicts that a single broad protrusion event would generate a large single spike of ERK activity, but smaller stochastic protrusions would generate multiple peaks of ERK activity. Indeed, when the authors stimulate serum-starved with a short pulse of EGF or a continued dose of EGF they see a large single spike or multiple spikes of ERK activity respectively. Finally, the authors determine the dose response of ERK activity to EGF stimulation is similar in conditions of low and high matrix stiffness, but find a higher peak response under high stiffness.

The main conclusion of the paper that protrusions drive ERK pulses is based almost entirely on correlative data and no mechanistic links are established. Since the data is largely correlative, further more direct experiments are necessary to draw such a conclusion. The core experiment using rapamycin to induce protrusive activities and measure ERK response are fairly indirect and have several issues as explained below that raise larger concerns about the conclusions the authors make.

Authors' reply: In this substantially revised manuscript we included new experiments and quantitative analyses to strengthen our conclusions and shed light on the molecular mechanisms, as outlined above and detailed below. For example, in addition to rapamycin-induced protrusions, we demonstrated that optogenetically triggered protrusions can also drive ERK activation. Moreover, we showed that FAK inhibition blocked protrusion-induced ERK activation, suggesting the role of FAK in mediating the feedback from cytoskeleton to Ras-ERK signaling.

Additionally, because previous studies have reported that ERK activity drives protrusion and migration rather than their hypothesis (Aoki et al 2017, Mendoza et al 2011, Chernyavsky et al 2005, and others), it is necessary for the authors to go beyond these studies and directly confirm their model that protrusion is not preceding but instead drives ERK activity. They need to put their work into the body of work from other labs that it is instead ERK activity that drives protrusion.

Authors' reply: As mentioned above, our new experiments using FRET-based EKAR showed little difference between the timing of protrusions and ERK activation. Therefore the causative relationship between ERK pulses and stochastic protrusions cannot be deduced from their temporal order. Rather, our conclusion is based on the following observations: (1) acute induction of protrusions drove ERK activation (Fig. 3g-i and Fig. 4a-c); (2) acute (~1 hour) inhibition of ERK signaling did not affect spontaneous and growth factor-stimulated protrusions (Fig. 4h-j) (note: although Mendoza et al. 2011 showed that MEK inhibition reduced EGF-triggered cell spreading, judging from their immunoblot and images the effect appears to reflect an increased baseline, possibly due to relief of a negative feedback from ERK to Ras, rather than a decreased response). These observations were strengthened by our new experiments and quantitative analyses. Therefore, we concluded that protrusions, rather than ERK, are the driver of

the pulsatile activation. We see no contradiction with earlier reports about the role of ERK in regulating protrusions and cell motility, as the time scale of the pulsatile phenomenon considered in our study is much faster (~15 min for individual pulses). We believe that the regulation of cell migration by ERK occurs at a much slower time scale. A paragraph was added in Discussion to address this important issue raised by the reviewer.

Further, the ERK KTR sensor the authors cannot measure localized ERK activity within the cell (only global activity). An ERK FRET sensor (such as one used in Aoki et al 2017) would provide critical spatiotemporal information on where ERK is activated relative to the Ras/PI3K activity in protrusions. Since the manuscript is correlative and does not provide significant novel mechanistic insights, it is not suitable for publication in Nature Communications.

Authors' reply: We carried out experiments using the FRET-based ERK sensor EKAR, similar to that used by Aoki et al. 2017. An increase in EKAR FRET ratio was associated with protrusions, but again the signal was not spatially localized (Fig. 1g, Supplementary Fig. 2c), apparently due to the fast diffusion of the biosensor ($D \sim 10\text{-}100 \text{ um}^2/\text{s}$ for typical cytosolic proteins) compared to the slow kinetics (minutes) of ERK activation. A membrane tethered version of the EKAR might solve the problem but to our knowledge is not yet available. As mentioned above, we have expanded on the mechanism about how protrusions drive ERK activation in the revised manuscript.

Specific Points:

1) In Figure 3 using the rapamycin-induced recruitment of Cdc25, iSH2, and Tiam1 the ERK activity is already increasing prior to the addition of rapamycin as shown in graphs in panels d, f, and h. This is very concerning and since these are the only experiment artificially inducing different protrusive activities and measuring the ERK response it doesn't strongly show what the authors claim. Did all individual cells show ERK increases prior to rapamycin, or is the heterogeneity of the 8 cells averaged creating this? Since this experiment is critical to the authors' main conclusion they must show this more convincingly.

Authors' reply: ERK activity always showed small (<4%) fluctuations in the baseline, but the increase induced by artificially triggered protrusions was much greater in magnitude (>20%). We have updated the analysis for Tiam1 CID by including more cells (n=19) from new experiments (Fig. 4b). In addition to CID, we also showed that ERK can be activated by optogenetically induced protrusions using PA-Rac1 (Fig. 4c).

2) In Figure 1f it is unclear why some Ras- peaks are called and others not, and also how ERK KTR peaks are defined since they vary in slope and size. The authors should more carefully explain this in the text. Also these plots allow for a more detailed analysis that was not sufficiently explored. What is the typical delay between a protrusive peak and an ERK spike? What are the distributions of delay times and can you predict the size or delay time of the ERK KTR peak from the size of the protrusive Ras peak?

Authors' reply: To eliminate the arbitrariness in the definition especially for small fluctuations in biosensor and cell boundary that constantly occur, in our new analyses we included all large, discrete ERK pulses responses, defined as over 20% change in

ERKKTR C/N ratio. In these cases the reversal of nuclear and cytoplasmic intensity of ERKKTR was evident, and the associated protrusions could be unambiguously identified. Among 12 such ERK pulses, 11 were preceded by discrete protrusions, whereas one showed diffuse but small cellular spreading. The timing between protrusions and ERK was analyzed by two different methods across cells: (1) cross-correlation between the protrusion area and ERKKTR (C/N) revealed a lag of 6 min (Fig. 1f); (2) the time to reach half-maximal peak for ERKKTR (C/N) was 5.02 ± 0.75 min (mean \pm S.E.M) behind that of protrusions (Supplementary Fig. 2d). We carried out similar analyses with the FRET reporter EKAR. Unexpectedly, we found no significant delay in EKAR signal increase and protrusions (Fig. 1h, Supplementary Fig. 2d). The discrepancy between the two biosensors was consistent with their activation kinetics upon EGF stimulation (Supplementary Fig. 2e). As explained above, the causative relationship between ERK and protrusions was subsequently established through perturbational studies.

3) Figure 1f and 1g, 4g and 4h, so show the relationship between ERKKTR and protrusions in just one cell. Since the conclusions of the paper rely on mostly correlative data like this, the authors should expand their analysis to multiple cells.

Author's reply: As stated above we have carried out quantitative analysis of the temporal relationship between ERK activation and protrusions across multiple cells (Fig. 1f, 1h, Supplementary Fig. 1d).

4) In Figure 2d it is unclear how this plot is generated. It is a single cell, what is the scale, should the axis have labels?

Authors' reply: The original figure 2d was not properly labeled. The image represents snapshots of protrusions identified by FDM in 6 different viewing fields for each of the five cell types. The original and FDM processed images are shown in Supplementary Video S8. We corrected the oversight and made changes to the text and legend to clarify the confusion.

5) What is the temporal limitation of the ERKKTR sensor? Have the authors compared the kinetics to available ERK FRET sensors? Since the authors conclude protrusions stimulate ERK activity it would be more convincing if they used a sensor with rapid representation of ERK activity. Experiments comparing dynamics of the ERKKTR to the FRET reporter would be very useful to address this.

Authors' reply: We carried out experiments using the FRET-based ERK biosensor EKAR. As pointed out above, we confirmed the association between protrusions and ERK activation but found no significant delay in EKAR signal increase (Fig. 1h, Supplementary Fig. 2d). The observation, while consistent with its faster kinetics compared to ERKKTR (Supplementary Fig. 2e), suggested that the timing of ERK pulses and protrusions did not reveal the causal relationship between these events. Instead, the capacity of protrusions to drive ERK activation was demonstrated by artificially induced protrusions using the chemical (Tiam1 CID) and optogenetic (PA-Rac1) approaches as described above.

6) In figures 1b, 1c, 2g, 4b, 4c, 4d, 4e, 4g, 4h and there is no colormap to indicate the enrichment levels of the sensors shown.

Authors' reply: We corrected the oversight by adding colormaps to the figures.

7) The stiffness of matrix used is vastly different in the western blot experiments and the ERKTR experiments shown in Figure 5 panels b, c, d, and e (12 vs 50 kPa). Does the KTR sensor not show difference in the conditions used in the western blot experiment? If not, what does the western blot of pERK show under these conditions?

Authors' reply: In our hands the differential responses to matrix stiffness is most noticeable in the 0.2~4 kPa range, and there is no significant difference between cells on 12 and 50 kPa substrates, both in pERK immunoblot and ERKTR imaging. Due to the long turnaround time for obtaining those soft substrate plates, our choice of specific stiffness was sometimes dictated by the materials available to us. Nevertheless, we included a new figure to show the responses of pERK to EGF for cells on 0.5 and 50 kPa substrates (Fig. 6f, g).

8) In figure 4h the imaging time for a short pulse of EGF is significantly shorter than the continuous EGF exposure shown in 4g. The authors should ensure there are not other delayed ERKTR activities or protrusions they are missing.

Authors' reply: We have increased the duration of the tracing after short stimulus as suggested by the reviewer (Fig. 5h). Our result confirmed that the increased ERK activation and protrusion generation after a short stimulus was transient.

Reviewer #3 (Remarks to the Author):

Yang et al. perform an analysis of live-cell reporter data to demonstrate a mechanistic linkage between Ras-mediated cytoskeletal activity and overall cellular ERK signaling patterns. Sporadic pulses of ERK activity have been observed in a number of studies, both in vivo and in vitro, but the origin of these kinetics is not well understood. The authors demonstrate that bursts of ERK activity coincide in time with cellular protrusions that are high in Ras activity, as judged by recruitment of Raf-RBD to the plasma membrane. They show that overall protrusive activity predicts ERK signaling activity across multiple cell lines, and also demonstrate that induction of actin signaling by engineered signaling molecules can evoke ERK activation. They develop a model of this connection as an excitable network, with a positive feedback at the level of actin polymerization driving a highly cooperative ERK response.

Overall this is an interesting development in understanding what shapes ERK activity kinetics within the cell, and these observations will be of interest to other researchers focusing on the systems biology of Ras and MAPK regulation. However, there are some important limitations of this work that make it difficult to recommend for publication in its current form:

1. The quantitative and statistical analyses of the phenomena presented are very limited. For example, a cross-correlation analysis would be an excellent way to characterize the data shown in figures 1F or 4G. It would also be very helpful to see some indication of the trends shown in

larger sample of cells, rather than just a single example in each case. Without more extensive analyses it is very difficult to judge the strength of these associations, and therefore to assess the importance of the mechanisms reported.

Authors' reply: We thank the reviewer for the excellent suggestion. We carried out cross-correlation analyses for protrusions vs. ERKKTR across multiple cells (Fig. 1f) and EKAR (Fig. 1h). The lag times obtained from cross-correlation are in line with those based on half-maximal intensity changes (Supplementary Fig. 2d). As mentioned above, we confirmed a lag between protrusions and nuclear exit of ERKKTR but found minimal delay in EKAR activation. The causative relationship between protrusions and ERK activation was therefore established through perturbation studies including synthetic activation of protrusions using CID and optogenetic approaches as well as pharmacological inhibition of different proteins as described above.

2. Correlations between signals (For example Figure 1F/G) are reported as relationships between peaks. Conceivably, peaks may not be the most important feature of the relationship between the signals. Comparisons of onset time, or rate of change, would also be very informative, especially given the importance of ordering of events for the model. Plots of the derivatives of the signals would be very helpful. Also, it is not always clear in which cases the frame difference method is being applied.

Authors' reply: We agree with the reviewer that the peaks may not be the best way to compare the timing of events, especially when the temporal profile is not symmetric or is noisy. The exact onset time is also hard to determine due to noise in the basal activity. In our new analyses the timing between protrusions and ERK activities was analyzed by two different methods that yielded comparable results: (1) cross-correlation between the protrusion area and ERKKTR (C/N) revealed a lag of 6 min (Fig. 1f); (2) the time to reach half-maximal peak for ERKKTR (C/N) was 5.02 ± 0.75 min (mean \pm S.E.M) behind that of protrusions (Supplementary Fig. 2d). We carried out similar analyses with the FRET reporter EKAR. Unexpectedly, we found no significant delay in EKAR signal increase and protrusions (Fig. 1h, Supplementary Fig. 2d). The discrepancy between the two biosensors was consistent with their activation kinetics upon EGF stimulation (Supplementary Fig. 2e). As explained above, the causative relationship between ERK and protrusions was subsequently established through perturbational studies. The frame difference method (FDM) essentially reflects the derivatives of the signals and is mainly used for the identification of protrusions. We have indicated when FDM is applied in figure legends.

3. The authors make a distinction between peripheral protrusion-related RBD/PH-Akt pulses that drive ERK activity and interior flashes that do not. While there is some discussion in Fig. S3 of how these differ, it is still not clear how this distinction is drawn from the data. A more comprehensive reporting of the data from multiple cells, as discussed above, would help a lot.

Authors' reply: We carried out cross-correlation analysis for ERKKTR vs. total or internal RBD signals from nine 2-hour videos. Our analysis showed a lack of temporal correlation between ERKKTR and either total or internal RBD signal (Supplementary Fig. 4e).

4. The mechanism underlying the kinetics shown remains somewhat obscure. The model described in Fig 4, for example, does little to suggest how specific molecular events create the positive feedback loops that are apparent in the data.

Authors' reply: We conducted additional experiments to shed light on the mechanism of protrusion-induced activation of ERK. First, inhibition of actin by latrunculin abolished phospho-FAK as well as paxillin patches (Supplementary Fig. 6a, b). Second, FAK inhibition caused dose-dependent reduction in pERK and pAKT (Supplementary Fig. 6c, d). Third, FAK inhibition blocked ERK activation induced by CID of Tiam1 (Fig. 4g). Together, these observations suggest that actin-driven protrusions activate FAK, which mediates the activation of Ras-ERK pathway. FAK has been shown to generate phosphotyrosine sites that recruit RasGEFs, leading to Ras activation (e.g. reviewed in Giancotti 1999). We included this pathway in the updated Fig. 3a and Fig. 6a.

One thing that is not yet clear is whether ERK is part of the positive feedback loop, although it is assumed to be downstream.

Authors' reply: We carried out additional experiments and quantification to demonstrate that MEK inhibitor did not affect EGF-stimulated (Fig. 4h) or spontaneous (Fig. 4i, j) protrusions. Therefore ERK is not involved in the dynamic generation of protrusions. In contrast, artificially triggered protrusions through CID and optogenetic approaches were able to trigger ERK activation, suggesting that protrusions are the driver for the pulsatile ERK activation. Note that our results did not exclude a possible role of ERK in regulating cell motility at a slower time scale as mentioned above and discussed in the revised manuscript (see Discussion).

The data from the PI3K and MEK inhibitor experiments are very confusing as presented. The PI3K inhibitor appears to have very different effects on ERK depending on the cell line used, and its effect on protrusions is hard to assess from the pictures shown in Fig. S4D - why not report protrusions/time as in Fig. S4C?

Authors' reply: The effect of PI3K inhibition was easier to see in cells with high basal level of protrusions and ERK activity, such as SKOV3 cells (Fig. 2b). The low basal activity of MCF7 made the effect of PI3K inhibition less pronounced. We discussed this point in the revised manuscript. PI3K inhibition leads to immediate loss of nearly all protrusions in SKOV3 cells accompanied by nuclear entry of ERKTR. We agree with the reviewer that the snapshots presented in our original submission failed to capture the dramatic change in protrusions. We therefore included a video (Supplementary Video S10) showing the responses of five cells to PI3K inhibition and added quantification on the basal cell surface area change (Supplementary Fig. 5b).

The effects of the MEK inhibitor on protrusion are also unclear - only a single cell is shown, in movie form; quantitation would be more helpful. Overall, the data presentation just seems sloppy on this point, and from the current presentation it is hard to see how strong conclusions can be drawn regarding the feedback loops.

Authors' reply: As stated above, our new experiments and quantification from multiple cells provided clear demonstration that MEK inhibitor did not affect protrusions (Fig. 4h,

i, j), whereas artificially triggered protrusions triggered ERK activation, indicating that protrusions are the driver for the pulsatile ERK activation.

Reviewers' comments:

Reviewer #2 (Remarks to the Author):

The reviewers have addressed most of my concerns and the conclusion of the manuscript are now much more convincing. The one point I am still concerned with is the specificity of the FAK inhibitor. In their overall model, this is now an important point and I hope that the authors can repeat the same experiments by knocking down FAK and/or by using a second independent type of inhibitor.

Reviewer #3 (Remarks to the Author):

In this revised manuscript by Yang et al, the authors have responded to the reviewers' concerns with additional data and more careful analysis. The main point of the paper - a linkage between protrusions with enhanced Ras and PI3K signaling activity - has been strengthened by the new analysis, and is one that will be of significant interest for the field. The investigation of the mechanism underlying this link is now more substantial, as the authors have now demonstrated that artificially induced protrusions are capable of inducing ERK activity, but naturally occurring protrusions are not dependent on ERK activity. The data now appear to be presented in a balanced way. Overall, this paper establishes a very interesting new connection between protrusions and growth factor signaling, and while there is undoubtedly much still to be learned about the specific linkage, publication of this paper at this point is appropriate and should lead to very interesting new work in the field.

Response to reviewer comments

Reviewer #2 (Remarks to the Author):

The reviewers have addressed most of my concerns and the conclusion of the manuscript are now much more convincing. The one point I am still concerned with is the specificity of the FAK inhibitor. In their overall model, this is now an important point and I hope that the authors can repeat the same experiments by knocking down FAK and/or by using a second independent type of inhibitor.

Authors' reply: Although activation of Ras/PI3K/ERK signaling by FAK has been demonstrated in various cellular contexts (e.g. reviews by Giancotti and Ruoslahti 1999, Sulzmaier et al. 2014), the crucial point in our study is that FAK mediates protrusion-induced activation of ERK, as shown by the inability of artificially triggered protrusions to activate ERK in cells treated with the FAK inhibitor PF-573228 (Fig. 4g). To address the reviewer's concern about inhibitor specificity, we carried out the same experiments using a second FAK inhibitor PF-562271. We found that PF-562271 robustly inhibited ERK activation triggered by CID of Tiam1 without blocking the response to EGF stimulation, thus confirming our previous finding (Supplementary Fig. 6e).

Reviewer #3 (Remarks to the Author):

In this revised manuscript by Yang et al, the authors have responded to the reviewers' concerns with additional data and more careful analysis. The main point of the paper - a linkage between protrusions with enhanced Ras and PI3K signaling activity - has been strengthened by the new analysis, and is one that will be of significant interest for the field. The investigation of the mechanism underlying this link is now more substantial, as the authors have now demonstrated that artificially induced protrusions are capable of inducing ERK activity, but naturally occurring protrusions are not dependent on ERK activity. The data now appear to be presented in a balanced way. Overall, this paper establishes a very interesting new connection between protrusions and growth factor signaling, and while there is undoubtedly much still to be learned about the specific linkage, publication of this paper at this point is appropriate and should lead to very interesting new work in the field.

Authors' reply: We are very grateful for the comments and suggestions from all reviewers, who helped improve our manuscript tremendously.

REVIEWERS' COMMENTS:

Reviewer #2 (Remarks to the Author):

The authors have addressed my concerns